# The NMDA receptor subunit GluN2D is a potential target for rapid antidepressant action

Stefan Vestring [1,2], Maxime Veleanu [1], Marina Conde Perez[3], Louise E. Schuberth [1], Martin Bronnec[1], Anna Li[1], Lovis M. Würz[1], Fatih Erdogdu[1], Julia Stocker[1], Johanna Moos[1], David Weigel[1], Alice Theiß[1], Elisabeth Wendler[1], Lotta M. Borger [1], Sabine Voita[1], Franziska Heynicke [1], Jakob Brandl [1], Fabian Hummel [1], Clotilde Vivet[1], Dorothea Jocher [1], Pauline Loewe[1], Simon Barmann[1], Lea Smoltczyk[1], Stella Zimmermann[1], Prejwal Prabhakaran [1], Granita Lokaj[3], David H. Sarrazin [4,5], Guillermo Suarez [1], Judith Bernhardt[1], Catherine du Vinage[1], Elisa Grießbach[1], Julia Lais[1], Nicole Gensch [6], Magdalena Wojtas[7], Shira Knafo [7,8,9], Jule Wendel[1], Jan Warneke[1], Jean-Paul Grohe[1], Stefan Günther [10], Aurélien F. A. Moumbock [10], Katharina Domschke [1,11], Tsvetan Serchov [1,4,5], Josef Bischofberger [12] & Claus Normann [1] ✉

Ketamine is the first glutamatergic agent in clinical use for major depression, but its primary target remains unclear. Further research is needed to develop more specific interventions with fewer side effects. Ketamine is a non-competitive antagonist of the glutamatergic N-methyl-D-aspartate (NMDA) receptor. Here, we show that ketamine preferentially targets GluN2D-containing NMDA receptors on interneurons, and that selective GluN2D antagonism is sufficient to produce rapid antidepressant-like effects. We use ketamine, the selective GluN2C/D inhibitor NAB-14, *Grin2d*-siRNA and chemogenetic approaches in hippocampal slices and in vivo mice. We find that GluN2D antagonism inhibits NMDAR currents in interneurons but not pyramidal cells, and that GluN2D-mediated recruitment of GABAergic inter-neurons controls inhibitory circuits regulating hippocampal activity and plasticity. In a mouse model of depression, GluN2D inhibition recovers excitation-inhibition balance, restores plasticity, and mimics antidepressant-like actions of ketamine with fewer side effects. These findings identify GluN2D as a highly specific target for novel antidepressant therapy.

Major depressive disorder (MDD) is a widespread and devastating condition that leads to enormous individual suffering and imposes a serious socioeconomic burden[1]. Despite its vast impact, current medical treatment options are limited. The delayed onset of the effects of classical antidepressants has been associated with increased suicidal ideation, which, in addition to the high percentage of patients who do not respond even after multiple treatment attempts, highlights the urgent need for novel and rapid-acting antidepressants. Subanesthetic doses of ketamine (KET) effectively reduce depressive symptoms within hours. An enantiomer of KET, *S*-ketamine, has been recently

approved for treatment-resistant MDD. However, the rapid onset and comparatively large effect size of KET treatment are often accompanied by distressing adverse effects limiting its broad clinical use, including dissociative symptoms, anxiety, increased blood pressure and the abuse liability of the party drug KET[2,3]. Moreover, the exact mechanism of action of KET is poorly understood.

According to the disinhibition hypothesis, KET is thought to preferentially bind to NMDA receptors (NMDARs) on GABAergic interneurons (INs), reducing the inhibitory input to pyramidal cells (PCs), which can increase glutamate release into the synaptic cleft or modify signal processing in the dendrites of postsynaptic neurons[4–7]. In addition, early findings demonstrated that the non-selective use-dependent NMDAR antagonist MK801 preferentially reduced the activity of GABAergic INs, leading to a delayed increase of the PC firing rate in the prefrontal cortex of awake rats[8]. Supporting the idea of increased glutamate release, increased extracellular levels of glutamate and enhanced glutamate cycling have been observed after KET administration[9,10]; probably due to increased gamma oscillations[11,12]. However, studies have also reported no change or a decrease in glutamate release after KET administration[13].

NMDARs are ligand-gated ion channels that are formed at most synapses by two ubiquitous GluN1 together with two GluN2(A-D) or GluN3(A-B) subunits as dimers of dimers or as co-assemblies of different subunits[14]. The subunit composition determines the distinct biophysical, pharmacological and functional properties of the NMDAR subtypes in the CNS[15]. The expression profiles of the four GluN2 subunits differ remarkably during development[16], in response to neuronal activity, between brain regions and cell types and in terms of their subcellular localization[15].

Here, we focused on the GluN2D subunit. In the embryonic brain, GluN2D subunits are highly expressed and decrease markedly during early postnatal development[16]. GluN2D subunit-containing NMDARs display lower single-channel conductance, sensitivity to $Mg^{2+}$ and $Ca^{2+}$ permeability, open probability and slower deactivation than GluN2A/B-containing NMDARs[17–19]. Notably, GluN2D subunits are predominantly expressed in hippocampal, cortical and cerebellar interneurons and are largely absent in principal neurons in adults[16,20–22]. GluN2D is expressed in 60-80% of somatostatin-expressing (SOM-INs) and parvalbumin-expressing interneurons (PV-INs) in adult mice[23,24]. Due to their unique expression pattern and electrophysiological properties, GluN2D-containing NMDARs on interneurons are expected to have a profound effect on hippocampal and cortical microcircuits, including the regulation of pyramidal neuron firing patterns, dendritic signal integration and synaptic plasticity[25–29]. Importantly, in a recent unbiased transcriptomic study in humans, *GRIN2D* gene expression was upregulated in depressed patients who subsequently responded to KET compared with non-responders, which could indicate an important role for GluN2D in the mechanism of action of KET[30]. However, available data on the regulation of GluN2D expression in animal models of depression and in depressed humans are sparse and partly contradictory[31–33]. This may be due to its generally low expression levels in the brain.

Here, we clarify the molecular mechanisms underlying the preferential inhibition of NMDARs on interneurons by KET and the resulting modulation of excitation-inhibition (E/I) balance and synaptic long-term potentiation (LTP), which is a missing link between NMDAR inhibition, synaptic network restoration and rapid antidepressant effects. We identify the GluN2D NMDAR subunit as a promising potential target for more effective, tolerable and selective antidepressant strategies.

## Results
### Differential inhibition of NMDA EPSCs in pyramidal cells and interneurons depends on the selectivity of antagonists for GluN2D
As a first step, NMDAR currents were evoked by extracellular stimulation in the hippocampal CA1 PCs and oriens-lacunosum/molecular

interneurons, which were identified by morphological and electrophysiological characteristics and fluorescence excitation, in SOM-Cre (SST tm2.1(cre)Zjh/J) td-Tomato mice (Fig. 1a). At a holding potential of +40 mV in the presence of the AMPAR antagonist CNQX, the non-specific NMDAR antagonist DAPV eliminated >90% of the remaining stable current (Supplementary Fig. 1a-c). To pharmacologically dissect the impact of GluN2D subunits on isolated NMDAR currents, we used NAB-14 and (R,S)-ketamine.

NAB-14 (an N-aryl benzamide, NAB) is a GluN2C/GluN2D-selective negative allosteric modulator[34]. As the GluN2C subunit is poorly expressed in adult mice[23], NAB-14 is selective for GluN2D in our experimental approach. Site-directed mutagenesis experiments indicated that NAB-14 binds to a site between subunits in the transmembrane domain, specifically interacting with the GluN2D M1 helix[34]. To gain insight into this interaction, we computationally docked NAB-14 against a published cryo-EM structure of GluN1A/GluN2D NMDARs (PDB ID: 7YFF)[35]. The putative binding pose indicates that NAB-14 forms pi-stacking interactions with W586 and F590 via its indole and benzene moieties, respectively. It also forms an H-bond with W637 as well as hydrophobic interactions with M589, L594, I633, and M634 (Fig. 1b). A recently reported cryo-EM structure of the TDM domain of the GluN1b/GluN2D NMDAR in complex with the allosteric GluN2C/GluN2D antagonist YY-23 confirmed the location of the proposed NAB-14 binding site[36]. Furthermore, using the same approach, we found that the binding pocket of the GluN2B-NAM Ifenprodil is clearly distinct from that of NAB-14. In hippocampal brain slices, the addition of NAB-14 to the bath solution dose-dependently reduced the amplitude of NMDAR currents in INs to a higher extend than in PCs (Fig. 1c, d). Increasing concentrations of NAB-14 inhibited NMDAR currents significantly in INs but not in PCs (Supplementary Fig. 1d,e). Other studies have shown that NAB-14 has lower inhibitory potency on triheteromeric GluN1A/2B/2D NMDARs than on GluN1A/2D NMDARs[37].

KET binds to the phencyclidine-binding site within the channel pore of the NMDAR. This results in a use-dependent open-channel block that decreases the mean channel open time. Furthermore, an allosteric closed block might decrease the open frequency, causing trapping of KET in the channel pore[38]. KET is rather unselective for GluN2 subunits; however, under physiological conditions with 1 mM $Mg^{2+}$ added to the bath solution, KET shows 5- to 10-fold selectivity for GluN2C- and GluN2D-containing receptors over GluN2A and GluN2B[17,39]. In our experiments, KET preferentially inhibited NMDAR currents in INs compared to PCs at most concentrations (Fig. 1e). Increasing concentrations of KET dose-dependently and significantly decreased the amplitude of NMDA EPSCs both in PCs and in INs (Supplementary Fig. 1f, g).

Negative allosteric modulators of GluN2B are currently under investigation as potential antidepressants[21]. To distinguish between the mechanisms of action of GluN2D and GluN2B inhibition, we assessed the modulation of NMDAR-mediated currents by different concentrations of the GluN2B-selective NAMs Ifenprodil and RO25-6981[40]. At a high concentration (100 μM), Ifenprodil significantly reduced NMDAR currents in PCs and INs compared to baseline, with no significant difference in effect between the two cell types (Supplementary Fig. S1h). RO25-6981 did not significantly affect NMDAR currents compared to baseline in either PCs or INs, and no significant differences were observed between cell types at any tested concentration (Supplementary Fig. S1i). This differential cell-type sensitivity clearly distinguishes the pharmacological profiles of GluN2D and GluN2B NAMs.

### Inhibition of GluN2D decreases the activity of feed-forward and feedback loops in hippocampal microcircuits
PCs are embedded in a complex microcircuit containing different populations of interneurons, which form inhibitory feedback loops (FBLs) and feed-forward loops (FFLs, Fig. 2a). Both loops end in

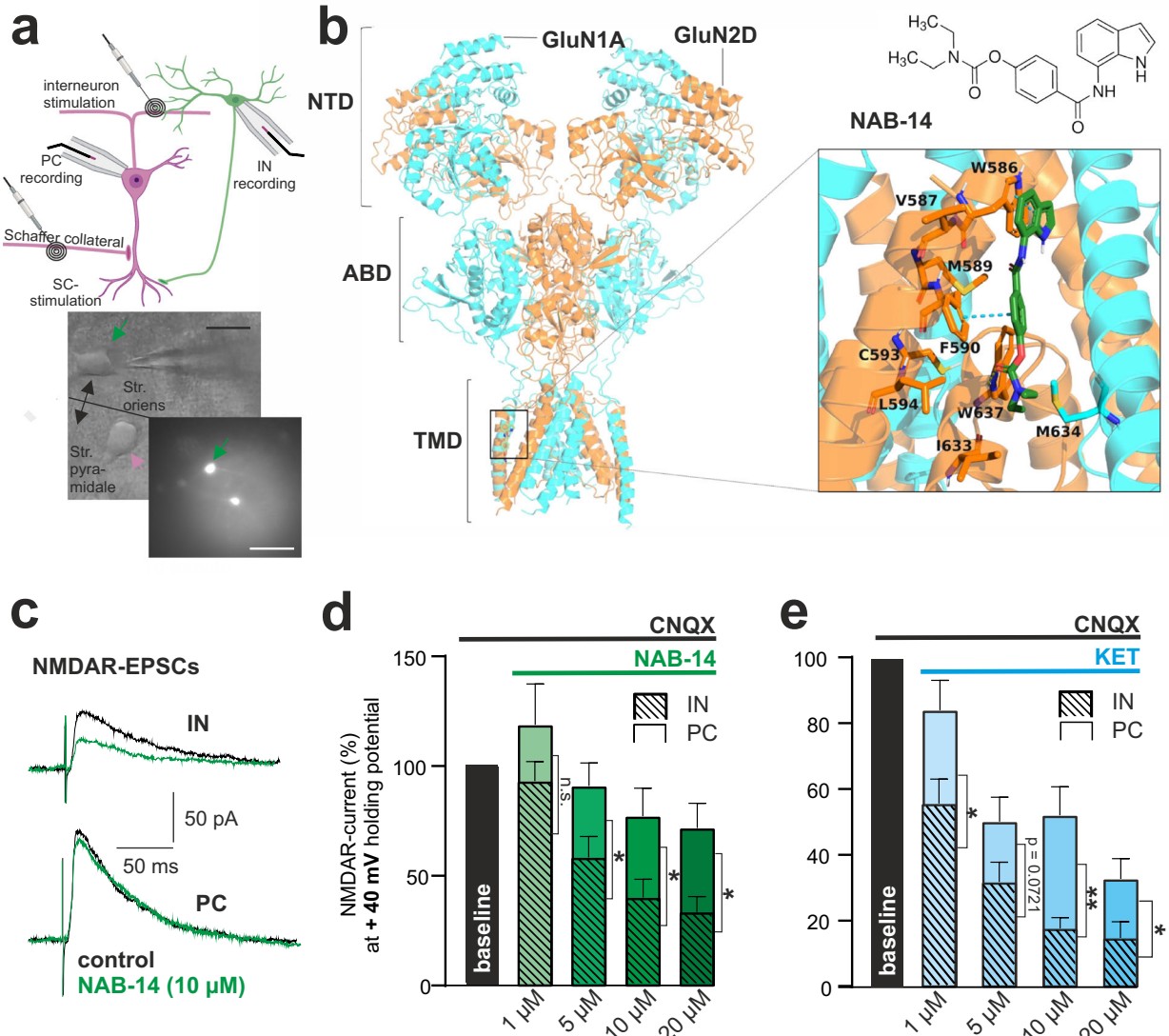

**Fig. 1 | Differential inhibition of NMDA EPSCs in pyramidal cells and interneurons depends on the selectivity of antagonists for GluN2D. a** Schematic overview of patch-clamp measurements in hippocampal brain slices. Synaptic activity was evoked in CA1 pyramidal cells (PCs, pink) by Schaffer collateral (SC) stimulation and in interneurons (IN, green) by extracellular stimulation in the stratum oriens. Representative image of SOM-INs (green) in wild-type C57Bl6 (native, scale bar: 20 μm) and SOM-Cre (SST tm2.1(cre)Zjh/J) td-Tomato mice (fluorescent, scale bar: 100 μm). **b** Left: Cryo-EM structure of the heterotetrameric GluN1a/GluN2D NMDAR (PDB ID: 7YFF), which is composed of two GluN1a chains (cyan) and two GluN2D chains (orange). It consists of three domains: the N-terminal domain (NTD), the agonist-binding domain (ABD), and the transmembrane domain (TMD). Right: 2D structure and docking model of NAB-14 (green) bound to GluN2D-TMD. The blue and yellow dotted lines represent pi-stacking interactions and H-bonds, respectively. **c** Representative NMDAR-EPSCs before (black) and after 10 μM

NAB-14 (green) in INs and PCs. **d** NAB-14 dose-dependently suppressed EPSCs more in INs ($n = 12$ [1/5/20 μM], $n = 7$ [10 μM]) than PCs ($n = 11$ [1/5/20 μM], $n = 7$ [10 μM]). Mixed-effects model (REML) with Geisser-Greenhouse correction found main effects of concentration (F = 15.82, $p < 0.0001$) and cell type (F = 8.33, $p = 0.0088$). Tukey's post hoc tests showed significantly stronger inhibition in INs at 5 μM ($p = 0.033$), 10 μM ($p = 0.036$), and 20 μM ($p = 0.010$). Two-sided tests. **e** Ketamine inhibited EPSCs more in INs ($n = 7$) than PCs ($n = 10$ [1/5/20 μM], $n = 8$ [10 μM]). Mixed-effects model (REML) with Geisser-Greenhouse correction showed main effects of concentration (F = 20.23, $p < 0.0001$) and cell type (F = 11.40, $p = 0.0042$). Tukey's post hoc tests indicated stronger IN inhibition at 1 μM ($p = 0.029$), 10 μM ($p = 0.005$), and 20 μM ($p = 0.038$). Two-sided tests. All data are shown as means ± SEM. Symbols: *$p < 0.05$, **$p < 0.01$. $n$ = number of cells. Source data are provided as a Source Data file. Created in BioRender. Vestring, S. (2025) https://BioRender.com/dpm4zwz.

GABAergic synapses at different locations of the PC dendrites, which contribute to dendritic signal processing[25,26]. GABAergic synapses of SOM-INs are located primarily at the distal dendrites of PCs and control synaptic NMDAR activation, burst firing, and synaptic plasticity[27]. In contrast, GABAergic synapses of PV-INs are near the soma of PCs, generating shunting inhibition and controlling the timing of spiking output[28].

The FFL was selectively activated by subthreshold Schaffer collateral stimulation, i.e., in the absence of PC action potentials (APs). The wash-in of 20 μM but not of 10 μM NAB-14 dose-dependently increased EPSP amplitudes in CA1-PCs (Fig. 2b). Under these recording

conditions, the increase in EPSPs is primarily driven by the inhibition of PV-INs by NAB-14. PV-INs express lower levels of GluN2D compared to SOM-INs[23]. This may account for the lower efficacy of 10 μM NAB-14 in this experiment relative to the direct measurement of NMDAR currents in stratum oriens (Fig. 1d), where SOM-INs predominate. In contrast, low- and very high-dose KET did not significantly modify the EPSP amplitudes (Fig. 2c). During the very-low-frequency stimulation of Schaffer collaterals performed in these experiments, which evoked single APs in FFL-INs and allowed a reset of the NMDAR to a closed state between each stimulation event, the open channel NMDAR blocker KET[38] had no effect. In contrast, the negative allosteric modulator NAB-

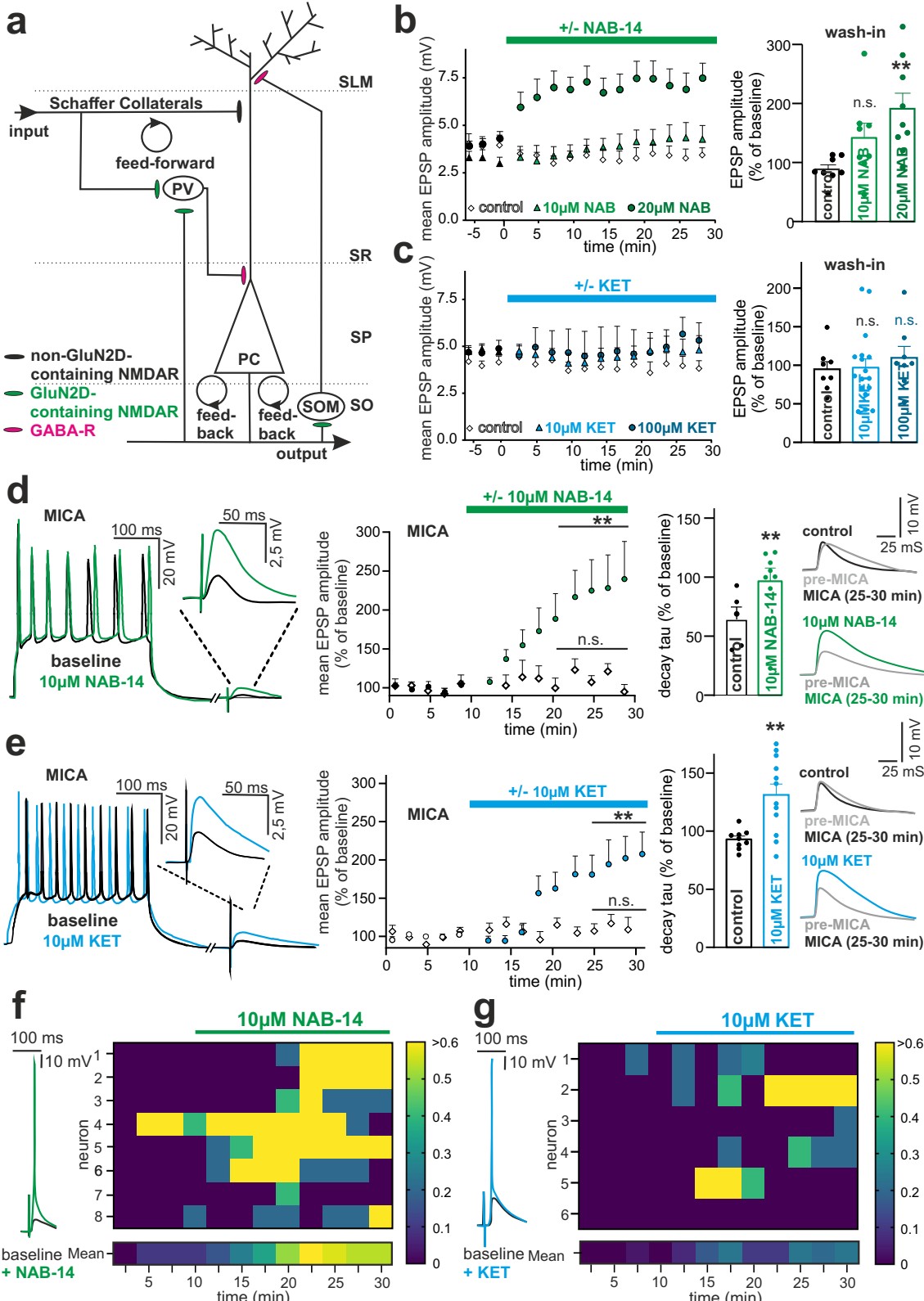

14 reduces NMDAR channel conductance independent of its opening state. The different mechanisms of NMDAR inhibition by the open channel blocker KET and the negative allosteric modulator NAB-14 might therefore explain the differential modulation of EPSP amplitudes by selective activation of the FFL in the absence of PC APs.

FBL-INs are highly dynamically recruited by excitatory PC activity and generate slow inhibitory postsynaptic currents (IPSCs) in PCs

mainly by activation of SOM-INs, effectively controlling PC NMDAR activation and synaptic plasticity[27,28,41]. To activate FBL-INs, we repeatedly injected a depolarizing current into the soma of CA1-PCs, evoking a burst of APs. Subsequently, after each AP burst, an EPSP was induced by Schaffer collateral stimulation, which served as a readout for the effects of the modulation of the excitation/inhibition (E/I) balance on glutamatergic transmission in PCs (microcircuit-activating

**Fig. 2 | Inhibition of GluN2D decreases the activity of feed-forward and feedback loops. a** Schematic overview of hippocampal feedback- and feedforward loops. PC, pyramidal cell. SOM, somatostatin-expressing interneurons; PV, parvalbumin-expressing interneurons; SLM, stratum lacunosum moleculare; SR, stratum radiatum; SP, stratum pyramidale; SO, stratum oriens. **b** (Left) Time course and (right) group analysis of normalized maximum EPSP amplitudes following wash-in of 10 μM NAB-14 (n = 8), 20 μM NAB-14 (n = 9), or vector control (n = 8). One-way ANOVA (F = 5.59, p = 0.011) with Dunnett's post hoc test showed a significant increase at 20 μM (p = 0.0056), not at 10 μM (p = 0.18), vs. baseline. **c** (Left) Time course and (right) group analysis of normalized maximum EPSPs after 10 μM (n = 17) or 100 μM (n = 8) KET, or vector control (n = 8). One-way ANOVA showed no effect (F = 0.318, p = 0.73); Dunnett's post hoc: p = 0.99 (10 μM), p = 0.69 (100 μM). **d** (Left) Representative microcircuit-activating (MICA) protocol voltage traces at baseline (black) and after 10 μM NAB-14 (green); insets show magnified EPSPs. (Middle) Normalized EPSP amplitudes increased significantly after NAB-14

(p = 0.0019, two-tailed paired t-test, n = 9), with no change in control (p = 0.51, n = 5). (Right) EPSP decay constant (τ) increased in NAB-14-treated cells (n = 8) vs. control (n = 5; p = 0.0071, two-tailed unpaired t-test). **e** (Left) Representative voltage traces at baseline (black) and after 10 μM KET (blue); insets show magnified EPSPs. (Middle) Normalized maximum EPSP amplitudes increased significantly after KET (p = 0.0019, two-tailed paired t-test, n = 13), with no change in control (p = 0.3289, n = 7). (Right) Decay constant (τ) increased in KET-treated cells (n = 12) vs. control (n = 9; p = 0.0028, two-tailed unpaired t-test). **f** NAB-14 converts subthreshold EPSPs to APs. (Left) Representative traces before (black) and after NAB-14 (green). (Right) Heatmap showing mean AP occurrence per 2.5 min across neurons over 30 min. **g** KET converts subthreshold EPSPs to APs. (Left) Representative traces before (black) and after KET (blue). (Right) Heatmap showing mean AP occurrence per 2.5 min across neurons over 30 min. All data are shown as means ± SEM. *p < 0.05, **p < 0.01. n = number of cells. Source data are provided as a Source Data file.

[MICA] protocol; Fig. 2d). After achieving a stable baseline for 10 min, NAB-14 was added to the bath solution, which increased the EPSP amplitudes and decay time constants compared to those of the control condition without NAB-14 (10 μM, Fig. 2d and Supplementary Fig. 2a,b). Similar effects were observed with KET (10 μM; Fig. 2e and Supplementary Fig. 2c). In contrast to the FFL-activating protocol (Fig. 2c), the MICA protocol induces a train of APs in PCs and INs so that the open channel blocker KET can effectively inhibit NMDAR currents in INs.

Coefficient of variation (CV) analysis of the EPSP slopes before and after the wash-in of NAB-14 was consistent with a predominant mechanism at the dendrites of postsynaptic neurons (Supplementary Fig. 2d). PC excitability, as assessed by the PC input resistance ($R_m$) and the number of APs during the burst, did not significantly change in the presence of either NAB-14 or KET (Supplementary Fig. 2e–h). We therefore concluded that GluN2D inhibition of interneurons shapes the amplitude and time course of synaptic transmission in PCs.

Next, we assessed how GluN2D inhibition in interneurons modulates excitatory activity in hippocampal PCs. Using a protocol similar to that used in the MICA experiments, Widman and McMahon demonstrated that KET increases the probability of synaptically driven AP[5]. In our experiments, NAB-14 and KET caused a pronounced conversion of EPSPs to APs at a fixed extracellular stimulation intensity (Fig. 2f–g and Supplementary Fig. 2i, j).

Taken together, these results suggest that GluN2D-mediated recruitment of GABAergic interneurons powerfully controls feedback and feedforward inhibitory circuits to moderate hippocampal network activity.

## GluN2D inhibition increases event-related excitatory network activity in the mPFC

To directly assess bulk network activity in vivo, we used a fiber-photometric approach. An optical fiber was implanted into the left hemisphere of the medial prefrontal cortex of Thy1-GCaMP6 mice. After three weeks of recovery, basal and event-related excitatory activity during exploration of a novel object was assessed (Fig. 3a). Neither KET nor NAB-14 (10 mg/kg i.p.) had effects on basal activity (Fig. 3b). However, KET and NAB-14 significantly increased bulk network activity during object exploration (event-related activity) as early as 20 min after application (Fig. 3c).

These results from the mPFC suggest that the microcircuit modulation by selective GluN2D inhibition, as demonstrated in the hippocampus as a model region, translates into increased event-related bulk excitatory network activity.

## Inhibition of GluN2D and SOM-IN activity increases hippocampal LTP

Since PV-INs modulate PC input at proximal dendrites, reduced PV-IN activity is concordant with an increase in EPSP amplitude and a conversion of subthreshold EPSPs to APs. SOM-INs preferentially activate

α5-subunit-containing GABA$_A$ receptors at CA1 PC dendrites and play a key role in dendritic computation and the shaping of LTP[27,29]. Moreover, the suppression of SOM-IN activity by KET has resulted in increased synaptically evoked calcium transients in the apical dendritic spines of pyramidal neurons in the mPFC[6,42]. Therefore, it is unclear whether GluN2D inhibition, especially by the partially selective antagonist KET, increases LTP induction in the hippocampus or potentially decreases LTP via direct inhibition of NMDA receptors in CA1 PCs[43,44].

We evaluated the effects of NAB-14 and KET on associative LTP induction in ex vivo hippocampal brain slices from mice. EPSPs evoked by Schaffer collateral stimulation were paired with postsynaptic APs in CA1 neurons. To avoid a putative ceiling effect of an optimized LTP induction protocol (Supplementary Fig. 3a, c)[45], we used a weak LTP protocol with 25 EPSP → AP pairings; this resulted in weak but significant LTP (weak-aLTP, Supplementary Fig. 3b, c). Bath application of 10 μM NAB-14 significantly increased weak-LTP (Fig. 4a). A maximal difference between the inhibition of NMDA EPSCs in INs and PCs was found at a concentration of 10 μM or higher (monoexponential fit, Fig. 4b, c.f. Fig. 1d). Lower concentrations of NAB-14 (1 and 5 μM) did not significantly affect either NMDA EPSCs in INs or weak-aLTP (Fig. 4a and Supplementary Fig. 3f). Bath application of 10 μM KET blocked weak-aLTP induction, whereas lower concentrations (1 and 5 μM) increased LTP (Fig. 4c and Supplementary Fig. 3g). This inverted U-shaped concentration–response relationship between KET and weak-aLTP reached its maximum at 5.3 μM KET (calculated maximum of Gaussian fit; Fig. 4d). At this concentration, the difference in inhibition of NMDAR excitatory postsynaptic currents (EPSCs) between INs and PCs (c.f. Fig. 1e) falls within the 5th percentile of the maximal observed difference (monoexponential fit, Fig. 4d). A similar inverted U-shaped dose-response curve has been reported for the behavioral effects of KET[46]. In addition, we tested the effect of KET on an optimized spike time-dependent LTP protocol. A total of 125 EPSP → AP pairings (aLTP) resulted in strong LTP (Supplementary Fig. 3a, c). Under these conditions, bath application of 10 μM KET had no effect on aLTP induction compared to the control (Supplementary Fig. 3d). The lack of inhibition of aLTP by 10 μM ketamine, in contrast to the complete suppression of weak-aLTP (Fig. 4c), can most likely be explained by the approximately 50% reduction in postsynaptic Ca$^{2+}$ influx at this concentration. Under strong stimulation conditions (125 EPSP→AP pairings), intracellular Ca$^{2+}$ levels remain sufficient to induce LTP. In contrast, during the weak stimulation protocol (25 EPSP→AP pairings) and in the presence of 10 μM KET, intracellular Ca$^{2+}$ does not reach the critical threshold required for LTP induction[47]. However, when inhibition was blocked by the GABA$_A$ antagonist picrotoxin (50 μM), the same concentration of KET inhibited aLTP (Supplementary Fig. 3e). Taken together, these results indicate that LTP is increased at concentrations of NAB-14 or KET at

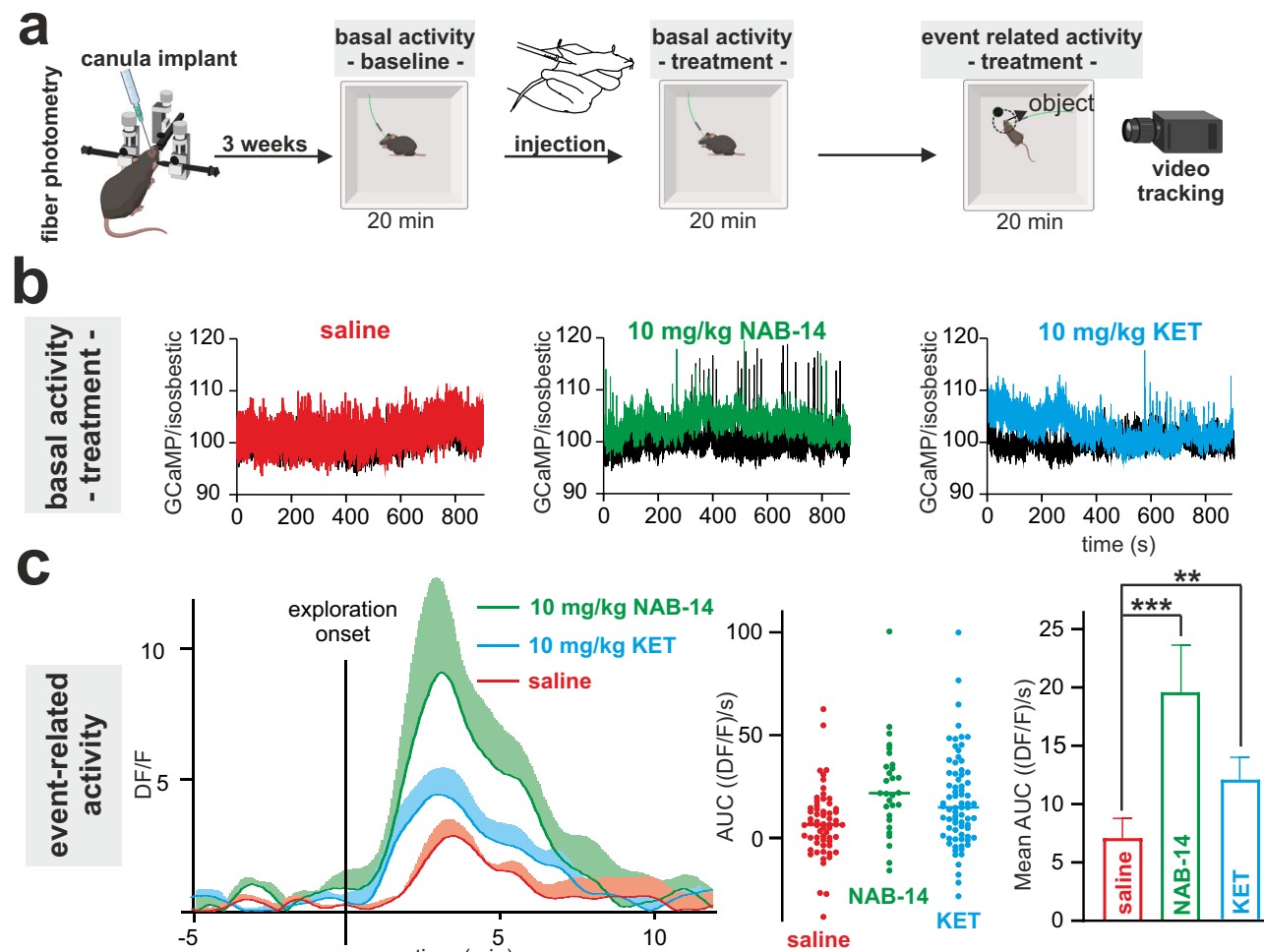

**Fig. 3 | GluN2D inhibition increases event-related excitatory network activity in the mPFC. a** Experimental procedures for fiber photometry experiments. **b** Basal bulk activity remained unchanged after the injection of saline (red), 10 mg/kg NAB-14 (green) or 10 mg/kg KET (blue); averaged time traces (n = 4). n=number of animals. **c** (Left) Mean ΔF/F changes over time during video-tracked object exploration in mice after injection of saline (red), KET (blue), or NAB-14 (green). (Middle) Individual data points and group means of the area under the curve (AUC; ΔF/F×sec) during repeated exploratory events. (Right) Summary group analysis. One-way ANOVA revealed a significant treatment effect (F = 8.41, $p = 0.0003$). Tukey's post-hoc tests showed significantly greater responses for NAB-14 vs. saline ($p = 0.0006$) and KET vs. saline ($p = 0.0084$), with no difference between NAB-14 and KET ($p = 0.28$). Data represent mean ± SEM from n = 29 (NAB-14), 75 (KET), and 61 (saline) event-related responses; collected from 12 mice total (4 per treatment group). **p < 0.01, ***p < 0.001. Source data are provided as a Source Data file. Created in BioRender. Vestring, S. (2025) https://BioRender.com/dpm4zwz.

which predominantly NMDAR EPSCs on INs are inhibited, but is blocked by high concentrations of KET, which inhibits NMDARs on PCs.

To assess the putative role of increased glutamate release in the modulation of LTP by NAB-14 or KET, we determined the paired-pulse ratio (PPR) before and after the induction of weak-aLTP. The PPR was unchanged in both the absence and presence of 10 μM NAB-14 or 5 or 10 μM KET (Supplementary Fig. 3f, g), indicating a postsynaptic effect. This suggests that an altered E/I balance and its computation in the dendrites of postsynaptic cells are sufficient for the modulation of synaptic plasticity and that a decisive role for a 'glutamate burst', as proposed for KET[7], is unlikely.

To directly assess the role of SOM-INs in LTP, we performed chemogenetic designer receptor exclusively activated by designer drugs (DREADD) experiments. By stereotactic injection, adeno-associated viruses (AAVs) were inserted into the hippocampus, leading to the expression of Cre-dependent excitatory (Gq) or inhibitory (Gi) DREADDs (pAAV5-hSyn-DIO-hM3D(Gq)/hM4D(Gi)-mCherry) in SOM-Cre mice (Fig. 5a, b). Activation of SOM-Gq-DREADDs with clozapine-N-oxide inhibited weak-aLTP induction, whereas significant

LTP could be induced with SOM-Gi-DREADD activation. LTP differed significantly between the two conditions (Fig. 5c). These results demonstrate that selective manipulation of SOM-IN activity modulates LTP induction in CA1 PCs.

### Inhibition of GluN2D restores LTP in an animal model of depression

Behavioral stress has been shown to increase GABAergic inhibition and compromise the excitation-inhibition balance[48-50], which might contribute to the well-known stress-related disruption of hippocampal synaptic plasticity[51-53]. SOM-INs might be specifically important for this process because they are strongly activated and increase inhibitory output during stress[48,54]. We investigated the modulation of LTP by repeated stress and GluN2D inhibition in a repeated stress (RS) animal model of depression, also referred to as the Chronic Behavioral Despair Model[55-57]. In RS, animals were subjected to ten-minute forced swim sessions on five consecutive days to induce a depression-like state, followed by a rest period of two days, and different readout assessments were performed afterward (Fig. 6a). In this model, the behavioral despair phenotype remains stable for at least 4 weeks in the

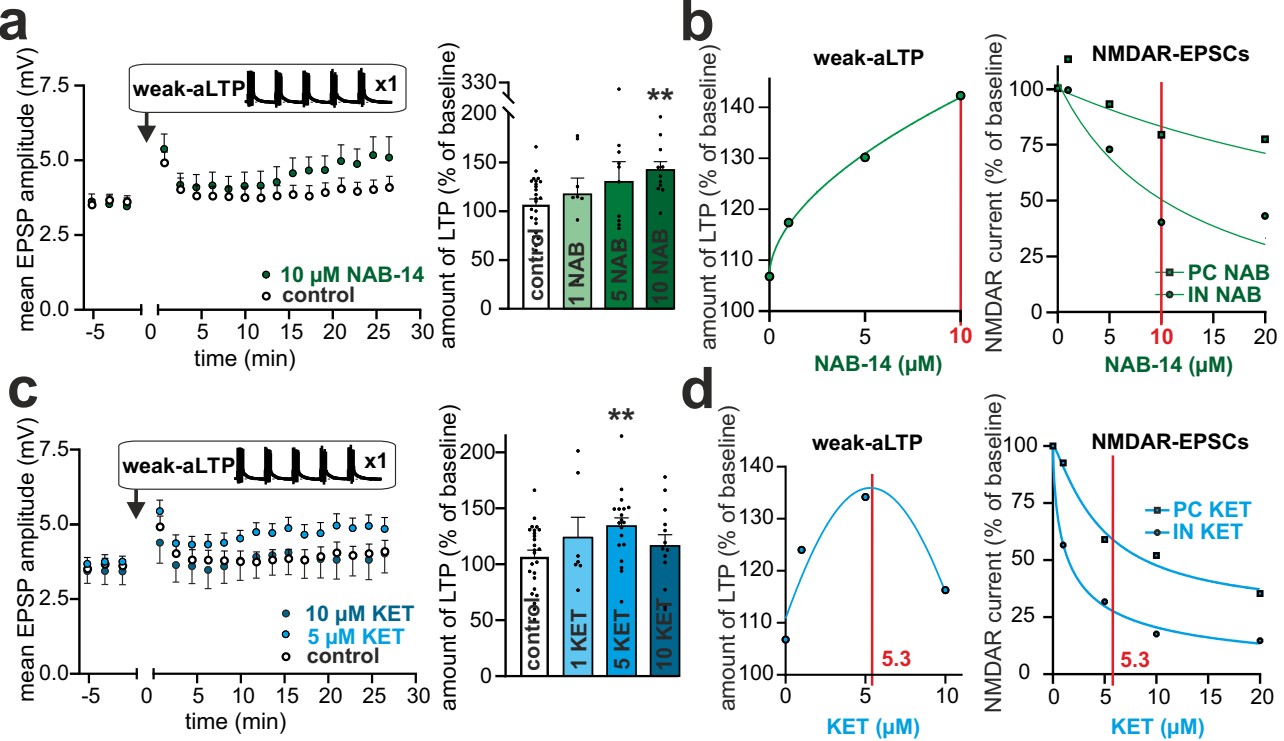

**Fig. 4 | Inhibition of GluN2D increases hippocampal LTP. a** Weak-aLTP was significantly enhanced by 10 μM NAB-14 ($n = 11$), while no effects were observed at 1 μM ($n = 8$) or 5 μM ($n = 12$) compared to control ($n = 25$). A Kruskal-Wallis test revealed a statistical trend toward a treatment effect (Kruskal-Wallis statistic=7.39, $p = 0.061$). Post hoc uncorrected Dunn's test indicated a significant increase at 10 μM NAB-14 compared to baseline ($p = 0.0066$), whereas neither 1 μM nor 5 μM NAB-14 significantly altered weak-aLTP ($p = 0.48$ and $p = 0.38$, respectively). **b** The addition of NAB-14 to the bath solution substantially increased the weak-aLTP in a monoexponential pattern (left). At 10 μM NAB-14 (red line), both LTP induction and differences in NMDAR-EPSCs between the IN and PC reached a maximum (monoexponential fit, right). **c** Ketamine modulated weak-aLTP in a non-linear, inverted U-shaped manner: no enhancement was observed at the low dose (1 μM, $n = 17$), a significant increase occurred at 5 μM ($n = 19$), and this effect was lost again at 10 μM ($n = 13$) compared to control ($n = 25$). A Kruskal–Wallis test revealed a significant treatment effect across groups (Kruskal-Wallis statistic=7.893, $p = 0.048$). Post hoc uncorrected Dunn's test confirmed a significant enhancement at 5 μM ketamine compared to baseline ($p = 0.0054$), while no significant changes were detected at 1 μM or 10 μM ($p = 0.57$ and $p = 0.42$, respectively). **d** The inducibility of weak-aLTP follows an inverted U-shaped relationship with the KET concentration added to the bathing solution (Gaussian fit). The maximum (red lines) weak-aLTP induction (135.9% of baseline) was calculated at 5.3 μM KET (left). Monoexponential fits to the decay of NMDAR-EPSCs in INs and PCs revealed that the difference in current inhibition between INs and PCs at 5.3 μM KET falls within the 5th percentile of the maximal observed difference (right). All data are presented as mean $n$ ± SEM. *$p < 0.05$, **$p < 0.01$. $n$ = number of cells. Source data are provided as a Source Data file.

absence of further interventions; chronic but not acute administration of tricyclic and selective serotonin reuptake inhibitor (SSRI) antidepressants has been found to reverse the depression-like phenotype[56].

LTP induction was completely abolished in ex vivo brain slices from mice injected with the control solution used to dissolve NAB-14 and sacrificed two days after the RS protocol. RS-induced LTP blockade could be reversed in a dose-dependent manner after intraperitoneal injection of 5 or 10 mg/kg NAB-14 (Fig. 6b and Supplementary Fig. 4a). KET injection (10 mg/kg) equally restored LTP induction (Fig. 6c).

As a nonpharmacological method for interfering with GluN2D, we used a small interfering RNA (siRNA) to achieve posttranslational gene silencing of *Grin2d*, the gene encoding GluN2D. We administered mouse *Grin2d*-siRNA (50 nM/animal) by intrathecal injection using in vivo jetPEI as a nonviral delivery vector. Injections were performed the day after the termination of the RS protocol; thereafter, the animals were left in their home cages for three more days (Supplementary Fig. 4b). Real-time PCR was used to verify that *Grin2d* RNA expression was downregulated in the hippocampus and frontal cortex (Supplementary Fig. 4c). The RS-induced decrease in LTP was fully reversed after siRNA treatment (Fig. 6d). Overall, the results of the RS-LTP experiments showed that KET and GluN2D inhibition reversed LTP deficits in an animal model of depression.

Additionally, we recorded spontaneous EPSP activity in hippocampal slices from naïve animals, following the RS protocol, and 24 hours after i.p. treatment of RS mice with 10 mg/kg KET or NAB-14. We observed a significant reduction in spontaneous excitatory activity after RS, which was restored by both NAB-14 and KET (Fig. 6e). These findings support the notion that GluN2D inhibition modulates the E/I balance in a way that facilitates excitatory network activity.

### GluN2D inhibition restores synaptic morphology after repeated stress

Functional alterations in plasticity are consolidated by morphological changes involving protein synthesis, especially increased spine density[58,59]. Therefore, we assessed the modulation of synaptic morphology after RS exposure and after GluN2D inhibition. Intraperitoneal injection of KET (10 mg/kg) numerically and NAB-14 (10 mg/kg) significantly increased spine density in biocytin filled PCs 24 hrs after RS (Fig. 7a, b). Although repeated stress did not significantly reduce spine density, this likely reflects the moderate and chronic nature of the applied paradigm, which induces subtle but functionally relevant synaptic alterations rather than pronounced structural remodeling. Markers for both excitatory glutamatergic presynaptic boutons (vGlu1) in the CA1 stratum pyramidale and inhibitory GABAergic presynaptic boutons (GAD 65) in the stratum moleculare, as assessed by immunostaining, were numerically reduced after the RS protocol.

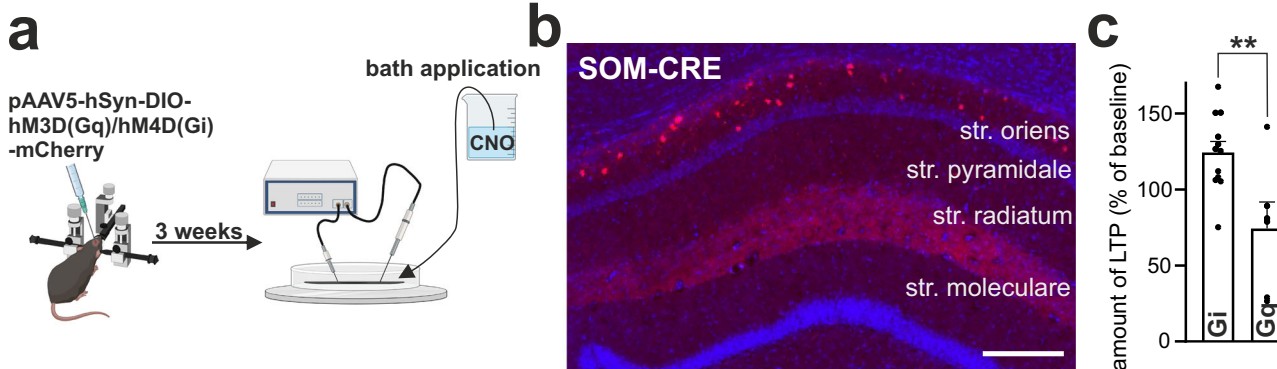

**Fig. 5 | Chemogenetic manipulation of SOM-INs bidirectionally modulates LTP.**
**a** Schematic representation of DREADD (designer receptor exclusively activated by designer drugs) experiments in SOM-Cre mice. CNO, clozapine-N-oxide.
**b** Fluorescence image of the CA1 region three weeks after DREADD virus injection in SOM-Cre mice showing the global expression of DREADD on SOM-INs (scale bar: 250 μm). This experiment was replicated once as proof of concept. Str., stratum.
**c** Activation of inhibitory DREADD (Gi) in SOM-INs by CNO (2.5 μM) resulted in an increase in weak-aLTP (two-tailed paired t-test, $p = 0.019$, $n = 12$), whereas activation of excitatory DREADD (Gq) on SOM-INs blocked weak aLTP induction (two-tailed paired t-test, $p = 0.093$, $n = 6$). Group comparison: Two-tailed unpaired t-test ($p = 0.0060$). All data are presented as mean ± SEM. **$p < 0.01$. $n$ = number of cells. Created in BioRender. Vestring, S. (2025) https://BioRender.com/dpm4zwz. Source data are provided as a Source Data file.

Compared with vector treatment, intraperitoneal injection of 10 μM NAB-14 24 h before tissue harvesting significantly increased vGlu1 and GAD 65 immunostaining after RS. KET treatment resulted in comparable but nonsignificant changes. (Fig. 7c–e). Western blotting was used to assess the levels of the postsynaptic density proteins PSD 95 and AMPARs (GluRA1). Two hours after i.p. application, both PSD95 and AMPAR concentrations were significantly greater in NAB-14-treated and KET-treated animals than in vector-treated animals subjected to RS (Fig. 7f–h) and Supplementary Fig. 5a, b). Taken together, these results indicate a trend toward the downregulation of both excitatory and inhibitory synapses, as well as AMPARs, following RS, and their plastic restoration by subsequent GluN2D inhibition.

Both LTP and spine growth are highly BDNF dependent. Protein kinase B/AKT and the extracellular signal-regulated kinase ERK are important signaling molecules downstream of the BDNF-TrkB receptor[60]. KET rapidly upregulates the phosphorylated and activated forms of both p-AKT and p-ERK[61]. Pharmacological inhibition of DUSP6, which is dephosphorylating ERK, could prolong KET's effect on synaptic plasticity, synaptogenesis, and behavior[62]. The activation of ERK is tightly linked to GluN2B[63,64]. To further differentiate the mechanism of action of GluN2D and GluN2B inhibition, we incubated hippocampal slices with NAB-14, KET and the GluN2B-NAM RO25-6981 for 45 min and assessed the level of phosphorylated AKT (p-AKT) and ERK1/2 (p-p44, p-p42) by Western blot. We found that NAB-14 significantly increased p-AKT, but not p-p44 and p-p42, whereas RO25-6981 increased p-p44 and p-p42, but not p-AKT. KET increased the phosphorylated forms of all molecules (Supplementary Fig. S5c–f). These results link GluN2D to AKT and confirms the coupling between GluN2B and ERK. It further supports that the mechanism of action of KET involves several targets.

### GluN2D inhibition has rapid antidepressant-like activity
Next, we assessed the behavioral effects of GluN2D inhibition. As a first step, we evaluated the basic pharmacokinetic properties and brain penetrance of NAB-14 following both intravenous (i.v.) and oral (p.o.) administration in male rats (Fig. 8a). After i.v. administration of 1 or 2 mg/kg NAB-14, we observed a short half-life of approximately 10 to 20 minutes in both plasma and brain, comparable to previously reported pharmacokinetic data for KET[65]. NAB-14 demonstrated efficient brain penetration, with a brain-to-plasma concentration ratio of approximately 0.75 at the two different doses (Fig. 8b). Oral administration of NAB-14 resulted in moderate absolute bioavailability (F = 12–52%) across two different formulations, which is substantially higher than that reported for KET[66]. Dose-dependent plasma concentrations were detectable for at least 6 h (Supplementary Fig. 6a, b).

In the RS animal model of depression, the immobility time, conceptualized as a measure of behavioral despair, gradually increased during the daily swim sessions, reaching a plateau at days 4-5, and remained stable in the delayed swim session on day 8 (Supplementary Fig. 6c). In previous experiments, we observed a persistent increase in immobility time for up to four weeks[56,57]. Thereafter, the immobility time returns to baseline levels without further intervention. 24 or 48 h after a single intraperitoneal injection of either 5 or 10 mg/kg NAB-14 or 10 mg/kg KET, as well as 72 h after the intrathecal administration of *Grin2d* siRNA, immobility time was significantly reduced compared to day 5 in the respective control condition. I.p. administration of saline had no effect (Fig. 8c).

As an alternative behavioral readout after RS, we performed the nose-poke sucrose preference test[67] (NP-SPT) using a progressive ratio reinforcement schedule to assess wanting-type anhedonia and reward behavior. Compared with naïve animals, RS mice showed a significantly reduced sucrose preference. Reward behavior was restored by a single injection of NAB-14 or KET and by the application of *Grin2d* siRNA but not by the application of a vector or scrambled siRNA (Fig. 8d).

We conclude that escape and reward behavior is compromised in an animal model of depression. Inhibition of GluN2D after RS normalized behavior to a level comparable to that of naïve animals; this indicates that GluN2D-mediated recruitment of inhibitory circuits powerfully controls depression-related behavior. GluN2D inhibition interferes with this process and has antidepressant-like activity in an animal model of depression.

### No effect of NAB-14 on spatial memory, motor coordination, locomotion or anxiety
The activity of hippocampal SOM-INs is necessary not only for depression-like behavior, but also for novelty-motivated spatial learning[68]. To assess the putative disruption of episodic memory in nonstressed animals, we used an object location memory task (OLT, Fig. 9a). First, two identical objects were presented without pharmacological intervention. There was no difference in the exploration of either object. Immediately thereafter, saline, NAB-14, or KET was injected, preceding a six-hour rest in the home cage, followed by a test session where one of the objects was presented at a different location. This newly located object was significantly more explored, with no difference between the treatment groups. We concluded that GluN2D

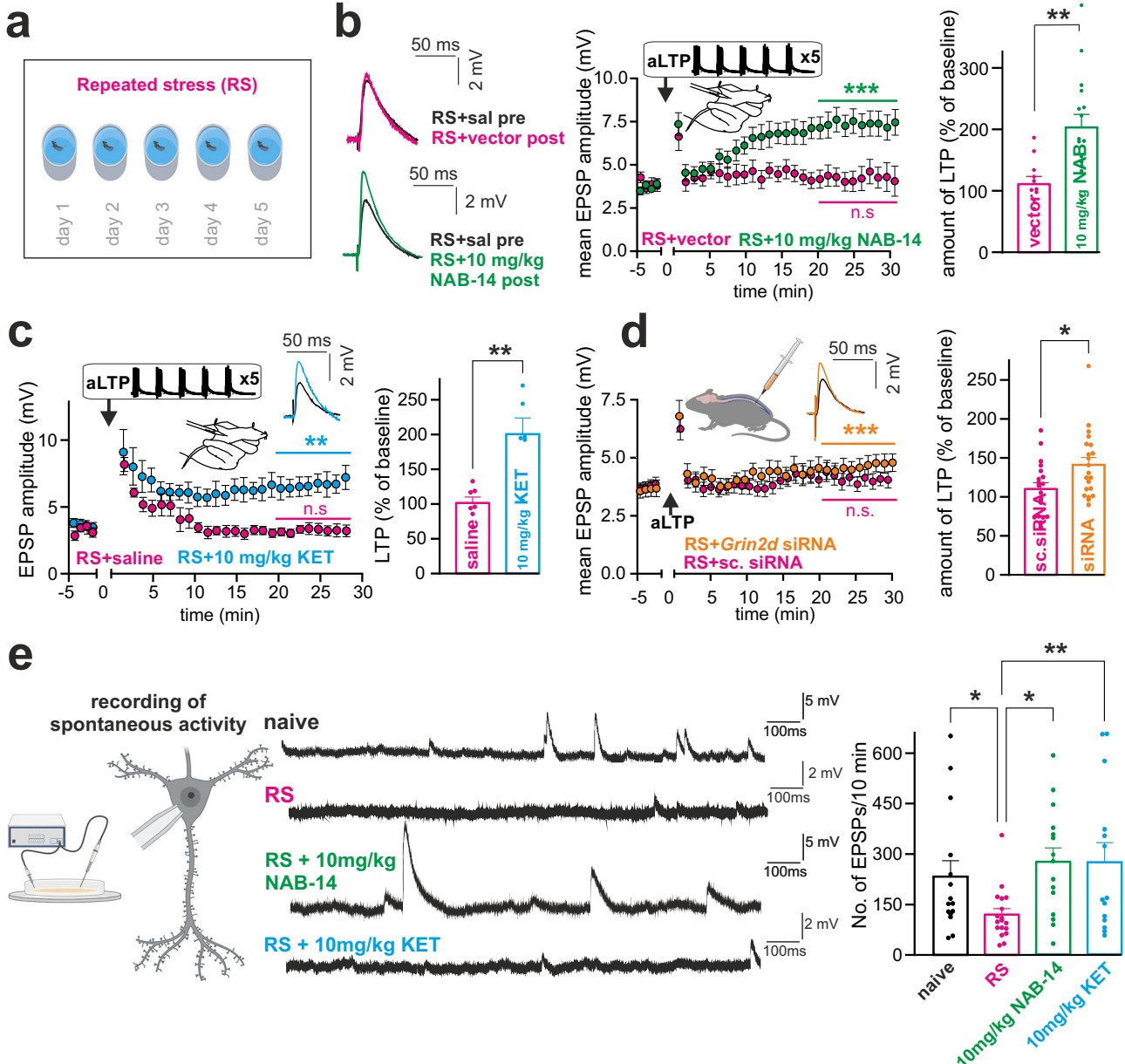

**Fig. 6 | Inhibition of GluN2D restores plasticity in an animal model of depression. a** Schematic overview of the Repeated Stress Model (RS). **b** After RS, no significant aLTP was induced (magenta; two-tailed paired t-test, $p = 0.4354$, $n = 11$), but a single NAB-14 injection (10 mg/kg, green; two-tailed paired t-test, $p = 0.0001$, $n = 16$) fully restored LTP. Exemplary EPSP traces (left), time course (middle) and group analysis (right) of the effect of NAB-14 on aLTP inducibility. Two-tailed unpaired t test control/vector vs. NAB-14 group, $p = 0.0028$. **c** A single KET injection (10 mg/kg, blue; two-tailed paired t-test, $p = 0.0045$, $n = 6$) resulted in a full rescue of RS-induced impaired aLTP (magenta; two-tailed paired t-test, $p = 0.9272$, $n = 7$). Exemplary EPSP traces (inset), time course (left) and group analysis (right). Two-tailed unpaired t-test control vs. KET group, $p = 0.0013$. **d** Treatment with intrathecal *Grin2d* siRNA (orange; two-tailed paired t-test, $p = 0.0002$, $n = 21$) but not with scrambled siRNA (magenta; two-tailed paired t-test, $p = 0.2858$, $n = 19$) reversed the RS-induced impairment of LTP. Exemplary EPSP traces (inset), time

course (left) and group analysis (right). Two-tailed Mann-Whitney test, $p = 0.0234$. **e** RS reduced spontaneous excitatory postsynaptic potentials (EPSPs), which were significantly restored by both KET and NAB-14. The experimental timeline is shown on the left, representative traces for each condition are presented in the center, and quantification of spontaneous EPSP frequency is displayed in the bar graph on the right. A Kruskal–Wallis test revealed a significant overall effect of treatment (Kruskal-Wallis statistic = 10.03, $p = 0.018$). Two-tailed post hoc Dunn's test indicated that RS ($n = 20$) significantly reduced EPSP frequency compared to naïve animals ($p = 0.039$, $n = 16$), and both KET (10 mg/kg, $n = 14$, $p = 0.026$) and NAB-14 (10 mg/kg, $n = 16$, $p = 0.0031$) reversed this effect. Data are shown as mean ± SEM. $n$ = number of cells. *$p < 0.05$, **$p < 0.01$, ***$p < 0.001$. Created in BioRender. Vestring, S. (2025) https://BioRender.com/dpm4zwz. Source data are provided as a Source Data file.

inhibition does not impair novelty-motivated spatial memory. Therefore, GluN2D inhibition might preferentially interfere with the exaggerated recruitment of inhibitory circuits during repeated stress, while the unperturbed AMPA receptor-mediated recruitment of these interneurons is sufficient to support episodic memory processing in naïve animals.

Finally, we tested for acute behavioral alterations following GluN2D inhibition alone. We assessed motor coordination with the rotarod test. After a single KET injection, the latency to fall was significantly shorter than that after NAB-14 or vector injection (Fig. 9b). This might be due to increased oscillatory activity in the basal ganglia induced by KET[69]. As GluN2D is not expressed in the striatum, these

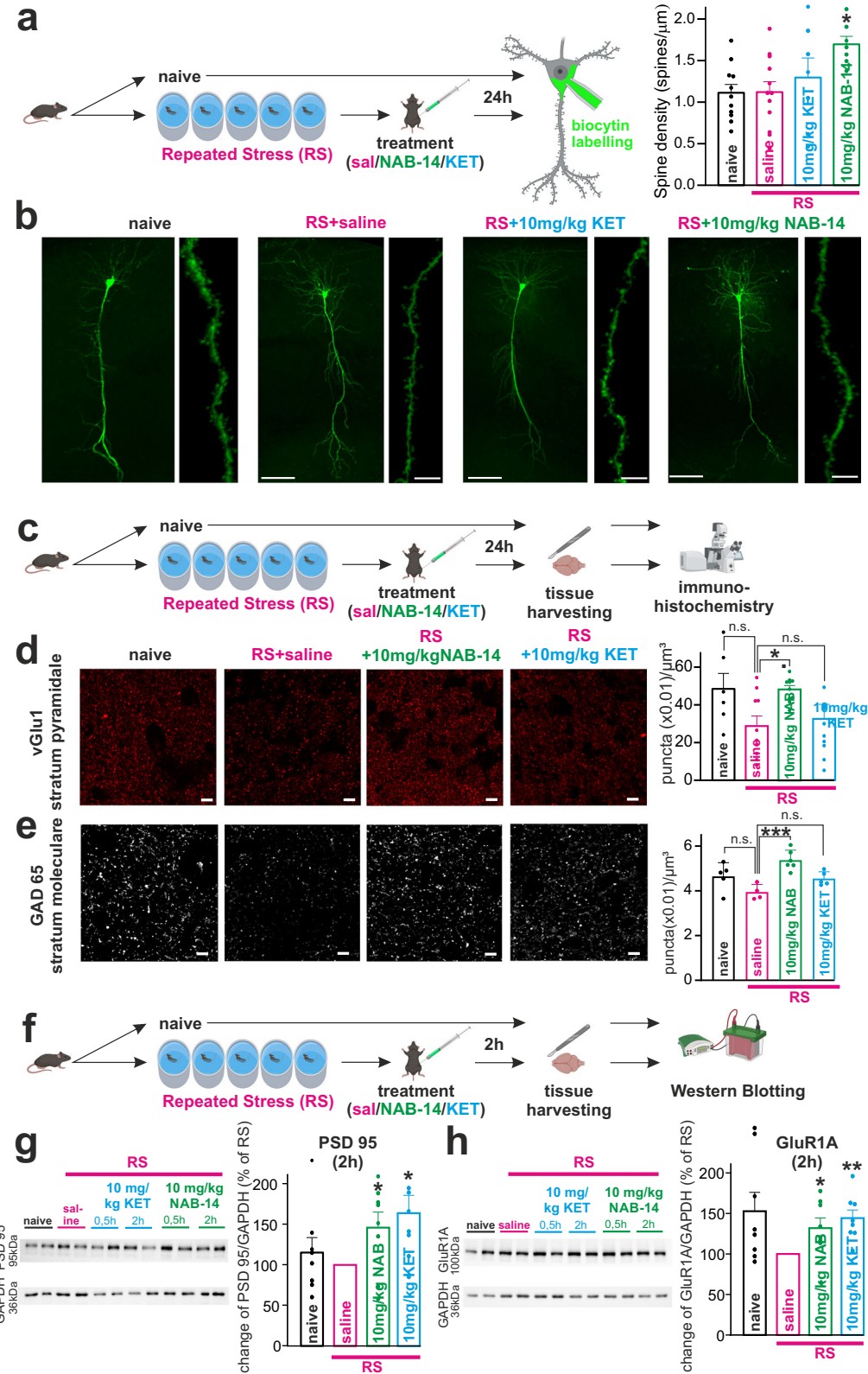

effects might be mediated by the action of KET on GluN2B receptors, which are abundantly expressed in striatal projection neurons[70]. Locomotion was unchanged after both NAB-14 and KET administration but decreased after administration of the benzodiazepine lorazepam, which was used as an active control (Fig. 9c). The time spent in the center in the open field test, which has been conceptualized as a measure of anxiety, was significantly decreased after KET but not

after NAB-14 injection in comparison to that after vector injection (Fig. 9d). Previous studies have reported acute anxiogenic effects of ketamine[71], whereas its anxiolytic properties at later time points—consistent with its antidepressant effects—are well documented[72]. In terms of motor coordination and anxiety-like behavior, the selective GluN2D antagonist NAB-14 produced fewer acute side effects than KET.

**Fig. 7 | GluN2D inhibition rescues stress-induced synaptic deficits. a** Workflow (left) and quantification (right) of biocytin-filled spine analysis. One-way ANOVA showed a treatment effect on spine density (F = 3.36, $p$ = 0.030). Dunnett's post-hoc test revealed increased density in NAB-14 vs. RS ($p$ = 0.021; $n$ = 7 vs. 12), with no differences for KET ($p$ = 0.73, $n$ = 7) or naive controls ($p$ > 0.99, $n$ = 11), n=number of cells. **b** Representative images of biocytin-filled (scale bar: 200 μM) and high-magnification dendritic segments (scale bar: 10 μM). **c** Immunohistochemistry procedures. **d** Representative vGlut1-immunostained boutons (stratum pyramidale, scale bars: 5 μm) under naive, RS, RS + NAB-14, and RS + KET conditions. Quantification of vGlut1 puncta revealed a significant treatment effect (Kruskal-Wallis, $p$ = 0.0054). Dunn's multiple comparisons test showed a significant increase in puncta density in the NAB-14-treated group ($n$ = 10) compared to RS ($p$ = 0.0248, $n$ = 10), no differences were observed for KET-treated mice ($p$ > 0.99, $n$ = 9) or naive controls ($p$ = 0.2599, $n$ = 6). $n$ = number of animals. **e** Representative GAD65-immunostained GABAergic boutons (stratum moleculare, scale bars: 5 μm) under naïve, RS, RS + NAB-14, and RS + KET. Quantification of puncta revealed a significant

effect of treatment (one-way ANOVA, F = 8.02, $p$ = 0.0017). Dunnett's post-hoc test showed an increase in puncta density in the NAB-14-treated group ($n$ = 6) compared to RS ($p$ = 0.0005, $n$ = 4), while no differences were observed for the KET-treated group ($p$ = 0.1582, $n$ = 5) or naive controls ($p$ = 0.0813, $n$ = 5). n=number of animals. **f** Western Blot (WB) procedures. **g** Group analyses of WBs. NAB-14 and KET treatment increased PSD95 in RS mice (two tailed-one sample t-test, normalized to RS group). Naive: $p$ = 0.441, KET: $p$ = 0.022, NAB-14: $p$ = 0.048, all n = 8. $n$ = number of animals. GAPDH (36 kDa, loading control) derived from the same experiments (gels and blots processed in parallel). **h** Group analyses of WBs. KET and NAB-14 treatment increased GluR1A in RS mice (two tailed-one sample t-test, normalized to RS group). Naive: $p$ = 0.058, KET: $p$ = 0.003, NAB-14: $p$ = 0.039, all n = 8. $n$ = number of animals. GAPDH (36 kDa, loading control) was blotted on the same membrane. Data are shown as mean $n$ ± SEM. $p$ < 0.05, **$p$ < 0.01, ***$p$ < 0.001. Created in BioRender. Vestring, S. (2025) https://BioRender.com/dpm4zwz. Source data are provided as a Source Data file.

## Discussion

The GluN2D subunit of NMDARs is unique due to its selective expression on interneurons in adulthood, distinct electrophysiological properties, and comparatively low overall expression levels. Selective GluN2D inhibition has profound behavioral consequences and might be a novel target for antidepressant action. Here, we describe a mechanistic sequence underlying the role of GluN2D-containing NMDARs in the stress response and the antidepressant-like action of GluN2D inhibition (Fig. 10).

Given the virtual absence of GluN2D expression in pyramidal neurons, the GluN2D inhibitor NAB-14 dose-dependently and selectively inhibits NMDAR currents in interneurons, whereas low-dose KET preferentially inhibits GluN2D subunit-containing NMDARs on GABAergic interneurons vs. NMDARs on PCs, depending on small dose-dependent differences in its ability to inhibit distinct NMDAR subunits. This might explain the small therapeutic window of KET for the treatment of MDD, as higher doses result in anesthesia but do not have antidepressant effects. Moreover, nonselective NMDAR inhibitors such as memantine do not exhibit antidepressant activity in rodent models or clinical trials[73]. We found that selective inhibition of GluN2D-containing NMDARs on interneurons by NAB-14 or in vivo application if Grin2d siRNA is sufficient to mimic many of the cellular and behavioral effects of KET. The different time scales of functional GluN2D inhibition by KET, NAB-14, and siRNA result in comparable behavioral effects, so GluN2D inhibition might be regarded as a common initial signal that is translated to sustained plasticity changes further downstream.

At the microcircuitry level in the hippocampus, GluN2D antagonism leads to an inhibition of feedback and feed-forward loops that end in GABAergic synapses at different locations of the PC dendrites. The feed-forward loop is dominated by fast-spiking perisomatic PV-INs, which effectively control spike timing in PCs[28,29,74,75]. SOM-INs in the feedback loop predominantly target nonlinear α5-containing GABA_A receptors (α5GABA_ARs) that generate slow inhibitory postsynaptic currents in hippocampal CA1 PCs[27,76]. Due to their strong outward rectification, α5GABA_ARs have a much larger conductance at depolarized membrane potentials, matching NMDAR properties, and therefore effectively control NMDA-dependent burst firing, dendritic integration and synaptic plasticity[27–29,77,78]. Both silencing of SOM-INs and a negative allosteric modulator of α5GABA_ARs increased subthreshold EPSPs and spiking output[27–29]. Interestingly, inhibition of SOM-IN synapses via negative modulation of α5 subunit-containing GABA_A receptors counteracts stress-induced disruption of the hippocampal E/I balance[50], showing some antidepressant-like effects similar to those of KET. Burst firing is particularly effective at promoting synaptic plasticity[79]. The activity of the feedback loop depends on AP firing in PCs as needed for LTP induction, which further emphasizes the predominant role of SOM-INs in the modulation of synaptic plasticity.

In our experiments, impaired activation of feed-forward and feedback loops by selective GluN2D inhibition shifted the E/I balance toward excitation, involving increased EPSP amplitudes and the conversion of EPSPs to APs. In most of our experiments, we used the hippocampus as a model region to dissect the cellular and microcircuit-level effects of GluN2D inhibition. A substantial body of literature supports the presence of significant structural and functional alterations in the hippocampus in both animal models of depression and patients with MDD[80]. Connectivity studies have further identified the hippocampus as a central hub within a dysregulated network in MDD, which includes the PFC, the amygdala, and the anterior cingulate cortex[81,82]. Given the widespread expression of GluN2D and the shared microarchitectural motifs such as feedback and feedforward loops across multiple brain regions, we do not propose that modulation of E/I balance in the hippocampus alone accounts for the behavioral effects of GluN2D inhibition. Rather, we suggest that these effects emerge from circuit-level interactions among distinct brain regions. In particular, interactions between hippocampal and prefrontal networks are well known to play a critical role in emotional memory and mood regulation[83,84]. Accordingly, we demonstrate that systemic administration of NAB-14 or ketamine leads to an increase in event-related bulk excitatory activity in the mPFC.

Selective GluN2D inhibition at doses equivalent to maximal NMDAR inhibition on interneurons is sufficient to increase hippocampal LTP. Conclusively, specific alterations in SOM-IN activity by inhibitory or excitatory DREADD receptors bidirectionally modulate LTP induction. It is highly probable that increased excitation and altered dendritic signal integration interact to augment LTP. Enhanced PC activity directly leads to an activity-dependent increase in protein synthesis, BDNF synthesis and release and mTOR signaling[85]. In turn, BDNF is necessary for the early and late stages of LTP and its consolidation by protein synthesis[60].

LTP is commonly considered the initial step in learning and memory formation and is consolidated by the formation of dendritic spines[58,59]. Behavioral stress as used in animal models of depression negatively modulates hippocampal synaptic plasticity, most likely through a corticosterone-dependent U-shaped relationship with mild stressors increasing and severe stressors impairing synaptic plasticity[86,87]. We confirmed that severe and subchronic stress induced by the RS protocol impaired LTP and synapse formation in the hippocampus.

Dysregulation of neuroplasticity is a key factor for both the pathogenesis and treatment of MDD[88,89]. Many forms of neuroplasticity are exquisitely sensitive to stress[90]. Psychosocial stressors, on the other hand, are highly relevant for the development of MDD in susceptible individuals[91]. LTP and other forms of neuroplasticity were found to be blocked in animal models of depression[52] and human MDD patients[92] and can be effectively restored by antidepressants[53,92,93]. The

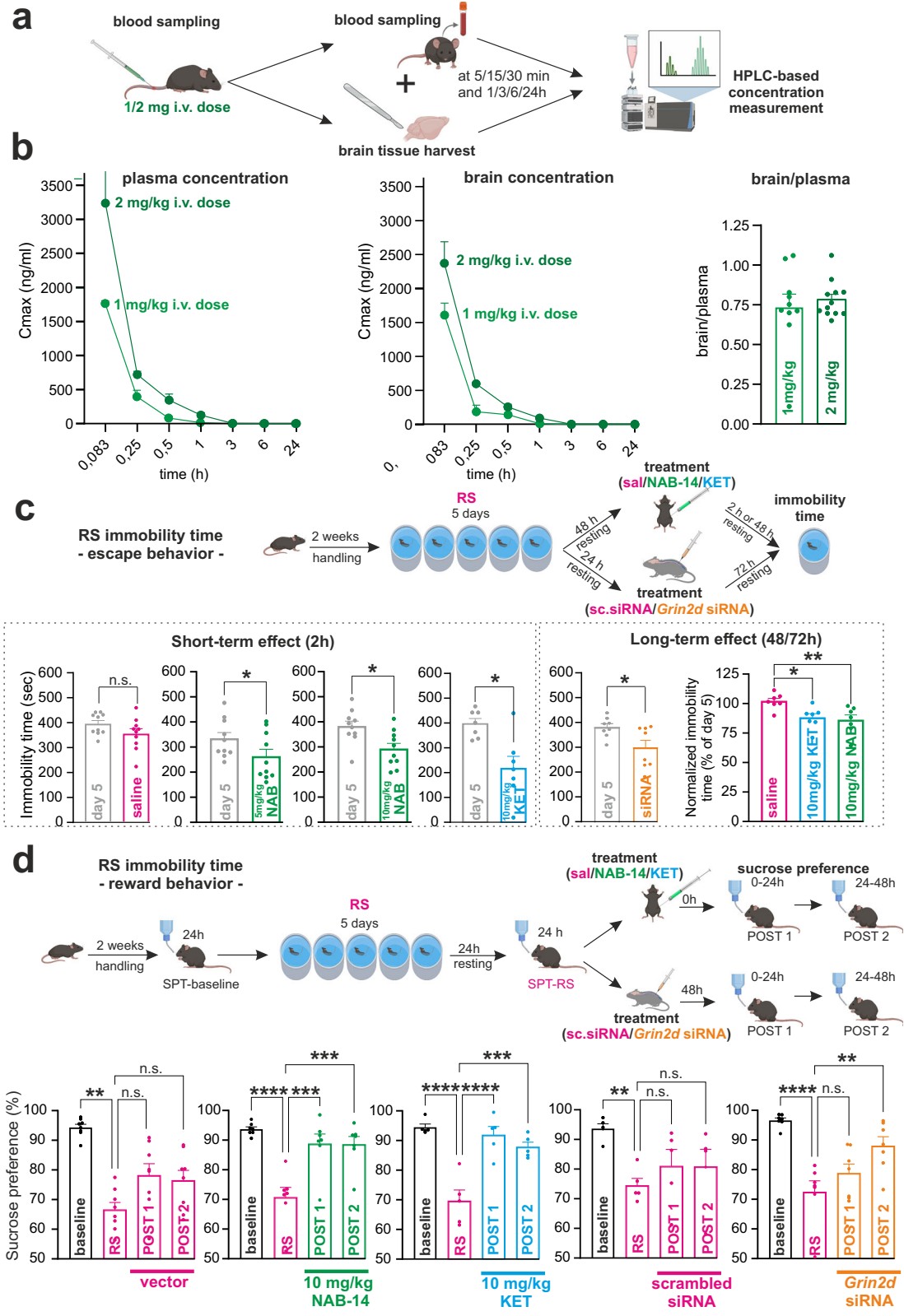

inhibition of LTP during sustained stress exposure might be adaptive to prevent an overload of negative valence learning and its neurobiological consequences. However, maladaptive downregulation of neuroplasticity could result in prolonged clinical depression, involving sustained shutdown of positive valence emotional learning.

In vivo GluN2D inhibition effectively reversed stress-induced impairments in LTP, synaptic morphology and density, as well as AMPAR expression, toward a naïve state. Restoration of impaired synaptic plasticity reestablishes neural adaptive capabilities and functionally reverses stress-induced network dysfunction. On a behavioral level, we found an impairment in escape and reward behavior after the application of the RS protocol, and these effects were reversed by GluN2D inhibition. However, disruption of learning by GluN2D inhibition is limited to negative valence experience, which is

**Fig. 8 | NAB-14 crosses the blood–brain barrier and GluN2D inhibition reverses stress-induced behavioral effects. a** Schematic representation of pharmacokinetic and brain penetration workflow after intravenous administration of 1 or 2 mg/kg NAB-14 in male rats. **b** (Left): Plasma concentration-time profiles following NAB-14 i.v. administration at 1 mg/kg (light green) and 2 mg/kg (dark green). (Middle): Corresponding NAB-14 concentrations in brain homogenates at the same time points. All $n = 3$ per dose and time point. N = number of animals. (Right): Brain-to-plasma concentration ratios at pooled time points 5/15/30/60 min. for 1 mg/kg ($n = 10$) and 2 mg/kg ($n = 12$) NAB-14 administration. N = number of animals. **c** (Top): Workflow for assessing escape behavior after repeated stress (RS)(Bottom): Short-term effects were tested 2 hours after i.p. administration of NAB-14 (5 or 10 mg/kg), KET (10 mg/kg), or saline. Two-tailed paired t-tests showed no effect for saline compared to values at day 5 of the RS protocol ($p = 0.091$, $n = 10$), but significant reductions in immobility time for NAB-14 (5 mg/kg: $p = 0.042$, $n = 10$; 10 mg/kg: $p = 0.030$, $n = 10$) and KET ($p = 0.032$, $n = 7$). Long-lasting effects were assessed 48 h after NAB-14 or KET administration and 72 h after *Grin2d* siRNA injection. *Grin2d*

siRNA reduced immobility at day 5 after RS (two-tailed paired t-test, $p = 0.019$, $n = 8$). One-way ANOVA across NAB-14, KET, and saline revealed a treatment effect (F = 5.95, $p = 0.0104$, all $n = 7$), with Dunnett's post hoc showing reduced immobility for KET ($p = 0.024$) and NAB-14 ($p = 0.010$) vs. saline. **d** (Top): Schematic overview of the procedures in the behavioral RS experiments assessing reward behavior. (Bottom): Treatment with NAB-14 (10 mg/kg), KET (10 mg/kg) and *Grin2d* siRNA reversed the stress-induced impairment of sucrose preference at 24 (POST 1) and 48 (POST 2) hours. Treatment with the vector used to dissolve NAB-14 and scrambled siRNA did not normalize sucrose preference. Repeated measures ANOVA, vector injection: F = 5.564, $p = 0.006$, $n = 6$; NAB-14: F = 13.02, $p < 0.0001$, $n = 7$; KET: F = 18.94, $p < 0.0001$; $n = 5$; scrambled siRNA F = 3.618, $p = 0.031$, $n = 6$; *Grin2d* siRNA F = 11.09, $p = 0.0002$, $n = 6$. Data are shown as mean ± SEM. N = number of animals. $p < 0.05$, **$p < 0.01$, ***$p < 0.001$, ****$p < 0.0001$. Created in BioRender. Vestring, S. (2025) https://BioRender.com/dpm4zwz. Source data are provided as a Source Data file.

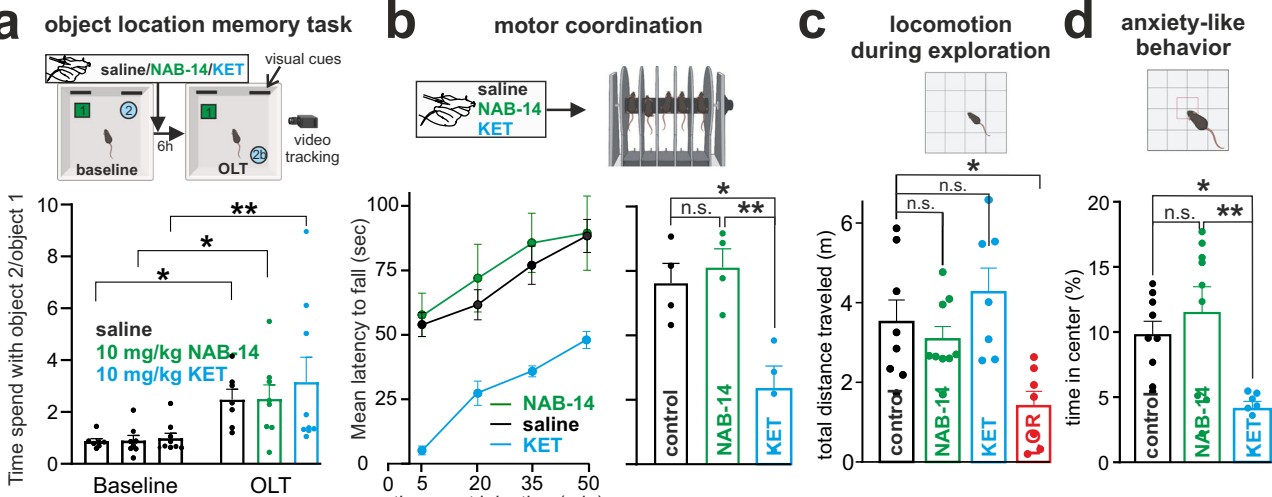

**Fig. 9 | No effect of NAB-14 on spatial memory, motor coordination, locomotion or anxiety-like behavior. a** Following training with two objects, mice received saline, 10 mg/kg NAB-14, or 10 mg/kg KET. Six hours later, one object was relocated (object location memory task, OLT). All groups showed increased exploration of the moved object, with no group differences. Two-way ANOVA: $p = 0.0003$ for time, 0.715 for treatment, $p = 0.805$ for time x treatment. Tukey's multiple comparison posttests, baseline vs. test session: saline $p = 0.044$, $n = 7$; NAB-14: $p = 0.033$, $n = 8$; KET: $p = 0.004$, $n = 9$. n=number of animals. **b** The latency to fall in the rotarod test was reduced by KET (10 mg/kg, blue, $n = 6$) compared to saline (black, $n = 6$) or NAB-14 (10 mg/kg, green, $n = 7$) injection. (Left): Two-way ANOVA: $p < 0.0001$ for time, $p = 0.002$ for treatment, 0.492 for trial × treatment. Post-hoc Tukey's test: vehicle vs. KET: $p = 0.0073$, vehicle vs. NAB: $p = 0.8571$, KET vs. NAB: $p = 0.0018$. n=number of animals (Right): Group analysis of the means of all time points per

group ($n = 4$, n represent the four time points). One-way ANOVA F = 10.25, $p = 0.0048$. Post-hoc Tukey's test: vehicle vs. KET: $p = 0.0137$, vehicle vs. NAB-14: $p = 0.8608$, KET vs. NAB-14: $p = 0.0062$. **c** Lorazepam treatment (LOR, red, 0.25 mg/kg, $n = 7$) decreased the distance traveled in the open field test, while KET (10 mg/kg, $n = 7$) and NAB-14 (10 mg/kg, $n = 9$) hat no effect vs. saline ($n = 8$). One-way ANOVA F = 6.163, $p = 0.0025$. n=number of animals. **d** The time spent in the center of the open field (as a percentage of the total time) was affected by KET (10 mg/kg, $n = 7$) treatment but not by NAB-14 (10 mg/kg, $n = 9$) treatment compared to saline ($n = 9$). One-way ANOVA F = 7.096, $p = 0.004$. Post-hoc Tukey's test: control vs. NAB-14: $p = 0.632$, control vs. KET $p = 0.027$, NAB-14 vs. KET: $p = 0.004$. $n$ = number of animals. Data are shown as mean ± SEM. *$p < 0.05$, **$p < 0.01$. Created in BioRender. Vestring, S. (2025) https://BioRender.com/dpm4zwz. Source data are provided as a Source Data file.

associated with enhanced recruitment and enhanced inhibitory output of GluN2D-expressing interneurons[48,54]. In contrast, we could not observe an effect during a neutral valence novelty-motivated learning task in nonstressed animals. This finding points to a central role for GluN2D and the recruitment of GluN2D-expressing interneurons in the stress response and the development of a depression-like phenotype.

Immediate changes in network function caused by disinhibition of interneurons and an increase in excitatory activity, as shown by increased EPSP amplitudes and conversion of EPSPs to APs following GluN2D inhibition, might underlie the very rapid behavioral effects of KET, which occur within minutes to hours. Subsequently, restoration of impaired synaptic plasticity reestablishes neural adaptive capabilities and functionally repairs stress-induced network dysfunction. It is highly probable that the downstream effects of FFL and FBL inhibition interact and are mutually reinforcing, e.g., increased LTP

inducibility and structural synaptic changes increase excitatory activity by upregulating AMPAR availability and vice versa. Functional alterations in plasticity are consolidated by morphological changes involving protein synthesis, especially increased spine density[85,94]. This might explain the sustained clinical effects of KET for several days after a single application or much longer after repeated applications[3,7,94,95]. Chronic application of conventional antidepressants for 2-3 weeks is needed to increase synaptic potentiation by more indirect mechanisms[94] or to preferentially inhibit long-term synaptic depression[55,92].

The behavioral consequences of pharmacological or siRNA-induced GluN2D antagonism in adult animals are in many respects opposite to those of constitutive GluN2D knockout[31]. GluN2D-KO mice displayed, e.g., increased immobility in the tail suspension task and reduced sucrose preference, in addition to reduced locomotion[31,96].

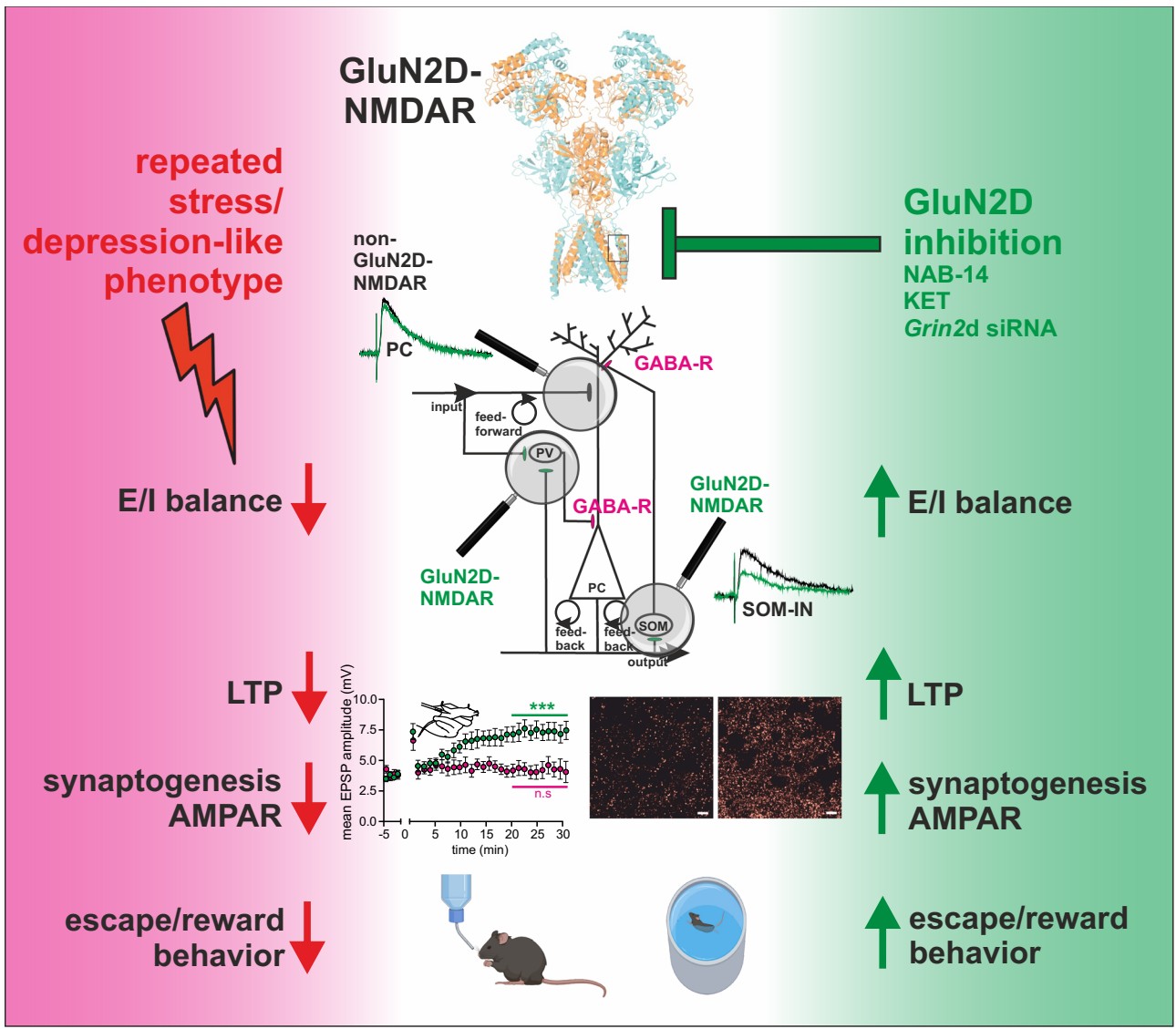

**Fig. 10 | Schematic representation of the mechanistic cascade linking GluN2D to the development and treatment of depression.** Left panel (rose background): Repeated stress induces a depressive-like state characterized by a shift in the excitation–inhibition (E/I) balance toward inhibition, impaired synaptogenesis, reduced AMPAR-mediated currents, and loss of associative long-term potentiation (LTP) in the hippocampus. Behaviorally, this is reflected by reduced escape responses and blunted reward sensitivity. Right panel (green background): Inhibition of GluN2D-containing NMDARs by ketamine, NAB-14, or *Grin2d* siRNA restores network function. In this recovered state, the E/I balance is normalized, synaptogenesis is enhanced, AMPAR currents are enhanced, and Schaffer collateral LTP is reinstated - paralleled by normalization of escape- and reward-related behaviors. Central upper panel: The cryo-EM structure of the heterotetrameric GluN1a/GluN2D NMDAR highlights the putative NAB-14 binding site within the receptor complex (see Fig. 1b for details). Central middle panel: A simplified diagram of the hippocampal microcircuit illustrates glutamatergic excitatory inputs onto pyramidal neurons and interneurons within feedforward and feedback loops (cf. Figure 2a). Magnified insets show that synapses onto interneurons are enriched in GluN2D-containing NMDARs, rendering them selectively responsive to ketamine, NAB-14, or *Grin2d* siRNA, whereas GluN2D-negative synapses are comparatively unaffected. Representative NMDAR current traces illustrate that NAB-14 reduces NMDAR-mediated EPSCs at GluN2D-positive synapses (green vs. black), but not at GluN2D-negative sites (cf. Fig. 1c). Central lower panel: Representative images show reduced synapse density following repeated stress and its reversal after GluN2D-targeted interventions (cf. Fig. 7d). Correspondingly, hippocampal recordings indicate that the stress-induced blockade of LTP is rescued by in vivo NAB-14 treatment (cf. Fig. 6b). The bottom line illustrates the normalization of escape and reward behavior following GluN2D modulation (cf. Fig. 8c, d), highlighting a multilevel restoration of structure, function, and behavior. Created in BioRender. Vestring, S. (2025). https://BioRender.com/dpm4zwz.

GluN2D expression decreases markedly during development and becomes localized to specific brain regions and cell types[16,21]. Tonic activation of GluN2D-containing NMDARs within the first weeks of postnatal development is thought to be critical for interneuron density and maturation and for GABAergic synapse density. Constitutive GluN2D mice are therefore expected to exhibit highly altered microcircuits in adulthood, which might explain these striking differences[97].

Here, we show that selective GluN2D antagonism alone is sufficient to induce rapid antidepressant-like effects. This notion has recently been supported by two independent studies reporting similar findings for the allosteric GluN2C/GluN2D antagonist YY-23 and the volatile anesthetic sevoflurane[36,98]. At least at lower doses, KET also preferentially inhibits NMDAR-mediated currents in interneurons, which may represent an important and convergent initial signal for modulating E/I balance, micro- and macrocircuit activity, synaptic function and morphology, and, ultimately, downstream behavioral outcomes. Our findings strongly suggest that the preferential inhibition of GluN2D-containing NMDARs on INs contributes to the rapid

antidepressant effects of ketamine. However, KET is undoubtedly a dirty drug with numerous additional targets that may amplify or complement its antidepressant mechanism[99,100].

Whereas the GluN2C subunit is virtually absent in adulthood, GluN2B is the second NMDAR subunit that is preferentially expressed in interneurons[16]. S-ketamine has been shown to bind to GluN2B-containing NMDARs[101], and GluN2B knockdown in somatostatin-expressing interneurons (SOM-INs) has been shown to reduce inter-neuron activity and prevent some behavioral effects of KET[6,42]. In our experiments, GluN2B NAMs were distinguished from NAB-14 by their lack of preferential inhibition of NMDA EPSCs in interneurons and by differential activation of the BDNF pathway components ERK and AKT. Consistent with our findings, the GluN2B-NAM RO25-6981 failed to elicit disinhibition of CA1 pyramidal neurons and was clearly distinct from the disinhibiting effects of KET[5]. In a number of preclinical and clinical studies, negative allosteric modulators of GluN2B showed rapid antidepressant activity comparable to that of KET, but also significant disruption of cognitive function, amnestic and dissociative effects[21].

We recently described a cholesterol-dependent binding site for KET and other antidepressants in the BDNF–TRKB receptor, which facilitates synaptic localization of TRKB and its activation by BDNF[102]. Use-dependent trapping of KET by NMDARs in the lateral habenula has been proposed to explain the sustained antidepressant action of KET despite its short elimination half-life[103]. Moreover, (2S,6S;2 R,6 R)-hydroxynorketamine (HNK), a major metabolite of KET, might exert an NMDAR-independent antidepressant-like effect through sustained activation of synaptic AMPARs[104]. Interestingly, a recent paper describes a loss of the sustained antidepressant-like effect of (2 R,6 R)-HNK in GluN2D-KO-mice[105]. However, the exact binding site of HNK is unknown, and its potential antidepressant efficacy remains controversial[106].

In this set of experiments, GluN2D-containing NMDARs were identified as promising targets for rapid-acting antidepressant-like action via rescue of the E/I balance and disturbed synaptic plasticity. Since these subunits are almost exclusively expressed on interneurons in the adult brain, they represent highly specific targets. Selectively modulating them may help avoid the adverse effects associated with less specific NMDAR antagonists, such as psychotomimetic and psychosis-like effects, abuse liability, anxiety, motor impairments, and anesthesia. Future studies will be required to fully evaluate the translational potential of GluN2D-selective NAMs through comprehensive behavioral characterization.

## Method
### Mice
All experimental procedures were approved by the relevant authorities in Germany (Regierungspräsidium Freiburg, TV-G-20-106, TV-G-20-87, TV-G-19-10, TV-G20-141, TV-G21-69, TV-G-22-116, TV-G-23-038, TV-G-24-033, TV-G-24-094) or France (CREMEAS, APAFIS n°2020042818477700), and conducted in accordance with EU Directive 2010/63/EU. Prior power calculations for behavioral assessments were performed to minimize the number of animals used.

### Rats
All pharmacokinetic studies were conducted in rats at Aptuit (Verona, Italy). All experimental procedures complied with the highest standards of animal welfare and were performed in accordance with the Italian Legislative Decree No. 26/2014 and European Directive No. 2010/63/EU. The studies were approved by the internal Aptuit Committee on Animal Research and Ethics and authorized by the Italian Ministry of Health (Project Authorization Code No. 35222). General procedures for animal care and housing followed the current recommendations of the Association for Assessment and Accreditation of Laboratory Animal Care (AAALAC).

## Animals and husbandry
### Mice
Wild-type C57Bl6N mice were obtained from Janvier (Le Genest-Saint-Isle, France) and Thy1-GCaMP6 mice from Jackson Laboratory (C57BL/6J-Tg(Thy1-GCaMP6f)GP5.5Dkim/J; Stock No: 024276/GP5.5). SOM-Cre (SST tm2.1(cre)Zjh/J) mice were provided by the Department of Biomedicine Basel and SOM-IRS-Cre/J from the Department of Physiology of the University of Freiburg. Animals were housed at the accredited facilities of the Universities of Freiburg and Strasbourg. Adult mice (10-14 weeks) were group-housed (up to five per cage) under standardized conditions (12 h light/dark cycle, controlled temperature (21 °C ± 2°) and humidity (55% ± 10%), food and water ad libitum, nesting material and enrichment).

All animals were handled by daily tunnel handling for 5 min over five days prior to behavioral testing to reduce stress. Welfare was monitored at least once daily by trained staff and additionally after invasive procedures, with health assessments including body weight, coat condition, locomotor activity, and posture. For stereotactic and other invasive procedures, anesthesia was induced and maintained with isoflurane, and peri-operative analgesia was provided (carprofen, buprenorphine). At the end of experiments, euthanasia was performed according to downstream analysis: for electrophysiology, mice were preoxygenated in 100% $O_2$ for 5 min before cervical dislocation and decapitation; for histology, mice were deeply anesthetized and transcardially perfused with fixative; for behavioral cohorts not undergoing perfusion or slice preparation, euthanasia was performed by cervical dislocation and decapitation without preoxygenation.

### Rats
For studies conducted in rats, male Sprague Dawley rats (Crl: CD(SD), 9 weeks old, 250–300 g) were bred and housed at Aptuit facilities in Verona, Italy. Animals were maintained in solid-bottomed plastic cages with sawdust bedding and either external watering systems or water bottles. Two to three rats of the same treatment group were housed per cage under controlled environmental conditions (temperature 20–22 °C; humidity 45–65%; 12 h light/dark cycle with species-appropriate illumination). Rats received a standard maintenance diet (Altromin 1324 IRR, rat diet) and filtered tap water *ad libitum*.

## Electrophysiology
Mice were preoxygenated in a 100% oxygen atmosphere for 5 min before cervical dislocation and decapitated according to national and institutional guidelines. Three hundred micrometer-thick transverse slices from the hippocampus were cut with a vibratome (VT1200, Leica Biosystems, Germany). Slices were prepared in artificial cerebrospinal fluid (aCSF) containing (in mM) 125 NaCl, 25 NaHCO3, 1.25 NaH2PO4, 2.5 KCl, 1 MgCl2, 27 glucose, and 2 CaCl2 (bubbled with carbogen (95% O2, 5% CO2)).

After 20 min of rest at 35 °C, the slices were kept at room temperature in aCSF, transferred to the recording chamber (volume of approximately 2-3 ml) and continuously superfused with aCSF (rate of approximately 5-10 ml/min-1). Differential interference contrast video microscopy (Zeiss Axioskop 2 FS plus, Zeiss Microscopy, Germany) was used to assess the location and morphology of CA1 pyramidal neurons. In addition to optical identification, neurons were classified according to their characteristic firing frequency adaptation to long depolarizing current pulses. Borosilicate glass tubes (2.0 mm outer diameter, 0.5 mm wall thickness; Hilgenberg, Germany) were used to pull the patch pipettes. The pipettes had an open resistance of 5-10 MΩ, and the series resistance (Rs) of 10-50 MΩ was compensated by a bridge balance. The patch pipettes were filled with two different internal solutions for current clamp (EPSP) and voltage clamp (current) measurements. The internal solution for the current clamp contained (in mM) 132 K-gluconate, 20 KCl, 2 MgCl2, 10 HEPES, 0.1 EGTA, 4

Na2ATP, and 0.3 NaGTP (pH adjusted to 7.2 with KOH). The internal solution for voltage clamp measurements contained 135 Cs-gluconate, 2 CsCl, 5 QX314, 10 HEPES, 10 EGTA, 2 MgCl2, 2 Na2ATP, and 2 TEA-Cl (pH adjusted to 7.2 with HCl). Osmolarity (280-300 mosmol/l) was controlled at the beginning of each experimental day with an osmometer (Osmomat, Gonotec, Germany).

A stimulation pipette (a patch pipette with an open resistance of 1-3 MΩ when filled with internal solution) was placed superficially in the stratum radiatum of the CA1 region, approximately 30–50 μm from the PC layer. Subthreshold EPSPs (for LTP protocols and wash-in experiments: 2-7 mV; for the microcircuit-activating (MICA) protocol: 5-10 mV) were evoked by Schaffer collateral stimulation with voltage pulses of 10–80 V (frequency of 0.1 Hz, duration of 200 μs) using a stimulus isolator (Model 2100 isolated pulse stimulator, A-M Systems, USA). The resting membrane potentials were between -75 and -65 mV, and the holding potential was -70 mV for PCs and between -60 mV and -65 mV for interneurons, with a holding potential of -60 mV. Hyperpolarizing voltage test pulses (50 ms/-5 mV) were applied to assess the input (Rm) and series (Rs) resistance after every 10th EPSP. For aLTP and weak aLTP, EPSPs were combined with postsynaptic action potentials (APs) triggered by short (3 ms) 700 pA current injections via a patch-clamp electrode. EPC-10 amplifiers (HEKA, Germany) were used, and the signals were filtered at 5 kHz. Patchmaster NEXT software (version 1.2, HEKA, Germany) was used for data acquisition and initial analysis. All experiments were performed at room temperature.

Experiments were discarded if (i) Rs changed by more than 30% during the experiment, (ii) evidence of ictal discharges was observed, (iii) the membrane potential between the start and end of the experiment differed by more than 5 mV, or (iv) the neurons did not respond to a firing pattern control pulse at the beginning and end of the experiments. Two experienced raters who were blinded to the experimental group agreed to the exclusion of the experiments. The experiments were performed in parallel; the experimenters were blinded to the treatment groups. A maximum of two hippocampal slices per animal were used. All groups included at least 6 animals with equal sex ratios.

### Identification of SOM-Ins

Slices from mice expressing tdTomato in SOM-INs (SOM-Cre (SST tm2.1(cre)Zjh/J)) or from wild-type mice were prepared as described above. Interneurons were visually identified by fluorescence imaging at an excitation wavelength of 553 nm and an emission wavelength of 580 nm by using a polychromator (Polychrome IV, Till Photonics, Germany), a fluorescence camera (Orca Flash 4.0, Hamamatsu, Japan), and the SmartLUX extension for Patchmaster NEXT software (HEKA, Germany). In slices from wild-type animals, SOM-INs were identified by their morphological characteristics, their location in the stratum oriens, and an initial membrane potential in the range of −60 mV.

### aLTP

Five EPSPs and five postsynaptic APs were paired at 100 Hz with a 5 ms delay (AP after EPSP). Five of these bursts of synchronized EPSP → AP pairs were applied at theta frequency (5 Hz), followed by an interval of 10 s and 4 more theta blocks, resulting in 125 EPSP/AP pairings.

**Weak-aLTP.** Unlike aLTP, only one theta block was applied, resulting in 25 EPSP/AP pairings.

### Wash-in experiments

EPSPs were elicited by Schaffer collateral stimulation at a frequency of 0.1 Hz. The mean baseline EPSP amplitude was calculated at 0-5 min. Substances were continuously applied to the bath solution after baseline recording for an additional 30 min. Mean EPSP amplitudes were calculated between 25 and 30 min after wash-in.

### Voltage-clamp recordings of NMDAR currents

At a holding potential of −70 mV (PCs) or -60 mV (INs), the cells were depolarized to +40 mV for 1 s. Five hundred milliseconds after the beginning of depolarization, extracellular stimulation was applied by Schaffer collateral stimulation (PCs) or by extracellular stimulation (≤ 10 V) in the stratum oriens at a distance of approximately 200 μm from the clamped interneuron (IN). Ten peak current amplitudes with an interval of 30 s were averaged before and after the bath application of substances.

### MICA

A 300 ms depolarizing current between +100 and +600 pA was injected via the patch pipette into a PC to elicit a train of 5–10 APs. After an interval of 600 ms, one EPSP (5–10 mV amplitude) was elicited by Schaffer collateral stimulation. This pattern was continuously repeated at a frequency of 0.2 Hz. Substances were added by bath application after a stable 10 min baseline recording for an additional 20 min.

### siRNA

Two hours after the induction phase of the RS protocol, the animals were anesthetized with isoflurane and placed on a heating pad. The lower back was shaved and disinfected, and the siRNA compound was slowly injected into the space between the L5 and L6 vertebrae. In some preparatory experiments, ink was injected to confirm the accuracy and reliability of the intrathecal injection technique (Supplementary Fig. 5a). The siRNA mixture contained (per animal) 0.06 μl in vivo-jetPEI solution (Polyplus-transfection, France), 50 nM siRNA, 5 μl glucose solution (10%) and 4.94 μl H2O, as described in a recently established protocol[107]. Afterward, the animals were allowed to rest for three days in their cages before the outcomes were assessed. The details of the siRNAs used were as follows: Ambion In Vivo Ready, HPLC-IVR (Thermo Fisher Scientific, s201423, catalog # 4457310); chromosome location: Chr. 7: 45831883-45872689 on GRCm38; RefSeq: NM_008172.2; translated protein: NP_032198.2; target exons: 2 and 3; and siRNA location: 1197. For the control experiments shown in Figs. 5d and 7 (scrambled siRNA), we used Ambion In Vivo Negative Control #1 siRNA (Thermo Fisher Scientific, catalog # 4457289). In all other siRNA experiments, the control group received the treatment described above, but the siRNA was omitted.

## DREADD and stereotactic surgery

Prior to surgery, 4 mg/kg carprofen and 0.1 mg/kg buprenorphine were injected subcutaneously. Mice were anesthetized with isoflurane (4% for sedation induction, 1.5% for maintenance) and placed on a stereotactic frame (51730UD, Stoelting, USA). Body temperature was maintained using a heating pad. A total of 300 nL of a viral suspension of either pAAV2/5-hSyn-DIO-hM4D(Gi)-mCherry (44362-AAV5, Addgene, USA) or pAAV2/5-hSyn-DIO-hM3D(Gq)-mCherry (44361-AAV5, Addgene, USA) was injected bilaterally into the hippocampus (-1.7 AP, +/-0.65 ML, -1.6 DV) of SOM-IRS-Cre/J mice using a 30 G syringe (Hamilton Neuros syringe) and a microinjection syringe pump (UMP3T, WPI, USA) at a rate of 200 nL/min. After injection, the syringe was left in place for five minutes before removal. After surgery, the mice received 0.1 mg/kg buprenorphine subcutaneously and 0.06 mg/100 ml buprenorphine in the drinking water overnight. The mice were then allowed to rest for 3 weeks before further experimental procedures.

## Fiber photometry
### Stereotactic surgery

Silica cannulas (400 μm core diameter, 0.37 NA, MFC_400/470-0.37_3.0 mm_MF2.5_FLT, Doric Lenses, Canada) were directly implanted in the left hemisphere of the prefrontal cortex (+1.7 AP, -0.3 ML and

-2.9 DV relative to bregma) of Thy1-GCaMP6 transgenic mice. After insertion, the cannula was fixed and protected using dental cement (Light Curing Nano Flowable Composite, Dline, Lithuania), solidified using a UV-light emitting device.

## Data acquisition and analysis

Fiber photometry allows for the measurement of neuronal calcium transients in real time[108,109]. The fiber photometry apparatus consisted of a Doric fiber photometry console (FPC, Doric Lenses, Canada), a 3-port fluorescent minicube with a built-in LED and detector (ilFMC3, Doric Lenses, Canada), and a USB DAQ board (USB-6210, National Instruments, USA). The minicube was connected to the cannula through a 5-meter low autofluorescence patch cord (400 μm core diameter, 0.37 NA, MFP_400/440/1100-0.37_5m_FCM-MF2.5_LAF, Doric Lenses, Canada). Two LEDs emitted light at 405 nm (isosbestic excitation wavelength) and 470 nm (GCaMP signal). The emission light was collected by the same optical fiber, passed through a GFP filter and reached the built-in photodetector. Data were then collected by the DAQ and a custom-written LabVIEW script (National Instruments, USA) at a sampling rate of 2 kHz. Data were then demodulated using a custom MATLAB (Mathworks, USA) script, and the GCaMP and isosbestic signals were analyzed using the Guppy fiberphotometry python suite[110]. For baseline recordings, the GCaMP channel was divided by the isosbestic channel and expressed as GCaMP/isosbestic. For the event-related recordings, the $\Delta F/F$ of the GCaMP/isosbestic signal was calculated (subtraction of the median of the recordings from the raw trace). To measure changes in GCaMP signals during behavior, mouse behavior was video tracked using DeepLabCut v2.3.5[111]. Timestamps were recorded each time the mouse entered a 35-pixel circular boundary around the object to explore. Peristimulus/event time histograms (PSTHs) were then computed using a window of 5 seconds before and 10 seconds after the event timestamp with baseline correction.

## Event-related activity

For the ERA experiment, other mice were temporarily removed from the home cage prior to the experiment. The test mouse cannula was plugged into the patch cord using a mating sleeve. LED excitation was then activated for a period of 15 minutes for habituation and to minimize initial photobleaching. Then, the camera and fiber photometry recordings were started. The mouse was allowed to move freely in its home cage for baseline recording. After 20 minutes of baseline, the drug (ketamine, NAB-14, or saline) was injected intraperitoneally, and the mouse was allowed to move freely in the home cage for another 20 minutes (after the injection period). Finally, an object was placed into the home cage, and the recording was continued for another 20 minutes.

## Biocytin filling and confocal microscopy

### Biocytin filling and fixation

Neurons were filled with biocytin during electrophysiological recordings[112]. Biocytin (10 mg/2 mL; Sigma, #B4261) was dissolved in internal solution by vortexing, followed by sonication for 10 min, and a final vortex. Neurons were injected with biocytin for 10–15 min while recording. To preserve cellular morphology, electrodes were carefully and rapidly withdrawn after injection. Following recordings, slices were fixed in 4% paraformaldehyde (PFA) in phosphate-buffered saline (PBS) for 24 h at 4 °C under gentle agitation.

### Alexa-Streptavidin labeling and mounting

Fixed slices were incubated in a blocking solution containing Alexa Fluor 488-conjugated streptavidin (1:500) for 48 h at 4 °C or over two days, with gentle agitation. Slices were then washed thoroughly in PBS at room temperature and mounted in ProLong Antifade Gold (Invitrogen).

### Blocking solution preparation

The blocking solution was prepared by dissolving 5 g sucrose and 2 g bovine serum albumin (BSA; Sigma) in 90 mL PBS, followed by the addition of 9 mL 10% Triton X-100. The solution was warmed to dissolve all components, aliquoted in 2 mL portions, and stored at −20 °C.

### Confocal microscopy

Biocytin-filled neurons were visualized using a confocal microscope Zeiss LSM800 equipped with an argon laser. Image stacks were acquired with a physical size of 89.4 × 89.4 μm and a logical size of 2100×2100 pixels. Each stack consisted of 20–50 image planes captured through a Plan-Apochromt 63× oil immersion lens (NA 1.4, working distance 180 μm, refractive index 1.515). To optimize resolution, a zoom factor of 1.5 was applied, resulting in a voxel size of 75.1 × 75.1 × 136.4 nm with a z-step of 0.15 μm. For each mouse, 1–3 randomly selected dendrites were scanned approximately 30-50 μm from the soma. The acquired stacks were processed using a 3D blind deconvolution algorithm (Imaris 9.7.2, Bitplane AG, Zurich, Switzerland).

### Spine density

Dendrites were traced using Imaris 9.7.2 software (Bitplane AG, Zurich, Switzerland) to quantify dendritic spine density[112,113]. Oblique apical dendrites from pyramidal neurons in the stratum radiatum of the CA1 region were traced from their proximal to distal tips. Dendritic spines were marked during the tracing process and considered as protrusions for inclusion in the analysis. Spine density was determined by counting all protrusions along the traced dendritic segments. This analysis was performed on 1–2 neurons per mouse, and no factors were applied to correct for spine counts.

### Spine morphology

Spine head volume was measured using Imaris 9.7.2 software (Bitplane AG, Zurich, Switzerland). For each dendritic segment, various intensity thresholds were applied to generate a model that was visualized as a solid surface using the IsoSurface module. The solid surface corresponding to the contour of each spine head was then selected. The three-dimensional image of each dendrite was rotated and carefully examined to ensure the accuracy of the surface selected for each spine head. Spines without visible heads, which were extremely rare, were excluded from the analysis.

## Immunohistochemistry

Brains were extracted after the intracardiac perfusion of mice with 20 mL of ice-cold phosphate-buffered saline (PBS; 8.1 mM Na2HPO4, 138 mM NaCl, 2.7 mM KCl and 1.47 mM KH2PO4 [pH 7.4]), followed by 15 mL of 4% paraformaldehyde (PFA) in PBS solution. The brains were postfixed in 4% PFA/PBS for 6 h at 4 °C and then cryoprotected for 48 h in a 30% sucrose/PBS solution at 4 °C. After sectioning with a cryostat, 30 μm-thick sections of the hippocampus were washed and blocked with 4% normal goat serum in PBS for 1 h at room temperature. The primary antibodies were diluted in PBS, 0.3% Triton X-100, and 1% goat serum and incubated overnight at 4 °C under agitation. The sections were then washed three times for five minutes in PBS supplemented with 1% Triton X-100 and incubated for one hour at room temperature with secondary antibodies diluted in PBS supplemented with 1% Triton X-100 and 1% goat serum. The sections were then incubated for 15 minutes at room temperature with nuclear 1 mg/mL 4,6-diamidino-2-phenylindole (DAPI, Thermo Fisher Scientific, USA, 1:1000) in PBS containing 0.2% Triton. After washing, the sections were mounted with Mowiol-DABCO (803456, Merck, Germany).

The following primary antibodies were used: guinea pig polyclonal anti-VGLUT1 (AB5905, 1:5000; Millipore, USA, lot #3878831) and mouse anti-GAD65 (ab26113, 1:500, Abcam, United Kingdom, lot #GR3308495-2). The following secondary antibodies were used: goat polyclonal anti-guinea pig Alexa Fluor 594 (Invitrogen, USA, A-11076,

2 µg/mL, lot #2540867) and goat polyclonal anti-mouse Alexa Fluor 488 (Invitrogen, USA, A-32723, 2 µg/mL; lot #XA336883). All images were taken using a Zeiss Celldiscoverer 7 microscope equipped with a confocal LSM 900 with AiryScan 2. The objective used was a Plan-Apochromat 50 x/1.2 W, N. A 1.2, H²O immersion. The laser intensity and gain were set to occupy the full dynamic range of the detector.

## Immunoblotting

### Ex vivo preparation of hippocampal slices for immunoblotting
Naive mice were preoxygenated with 100% oxygen for 8 minutes and euthanized by cervical dislocation. Brains were rapidly extracted and placed in carbogenated artificial cerebrospinal fluid (aCSF). The cerebellum was removed to create a flat surface, and the brain was mounted bulbs-up onto a vibratome carrier using water-activated adhesive. Sectioning was performed at 300 µm thickness with a cutting speed of 0.40 mm/s; the blade was raised and retracted after each cut to minimize tissue damage. Slices were immediately transferred into carbogenated aCSF for incubation.

Slices were incubated in aCSF containing either ketamine-hydrochloride (10 µM), NAB-14 (10 µM), or Ro 25-6981 maleate (3 µM). Each compound was initially dissolved in DMSO to prepare concentrated stock solutions before dilution in aCSF. Control slices were incubated in aCSF containing 0.1% DMSO to ensure equal solvent concentrations across all treatment conditions.

### Immunoblotting
Dissected brain regions or whole acute slices were mechanically homogenized in either homogenization buffer (320 mM sucrose, 4 mM HEPES [pH 7.4], 2 mM EDTA) or immunoprecipitation (IP) buffer (50 mM Tris HCl [pH 7.4], 120 mM NaCl, 5 mM EDTA, 0.5% Triton X-100) and centrifuged at $800 \times g$ for 10 min at 4 °C. All buffers contained a phosphatase and protease inhibitor cocktail (Sigma–Aldrich, USA). The protein concentrations were determined using a BCA assay kit (Thermo Fisher Scientific, USA) according to the manufacturer's instructions. DTT (10 mM) and bromophenol blue were added to total (mixed 1:1 with lysis buffer) lysates, boiled for 10 min at 95 °C, separated by SDS–PAGE on 7.5–12% acrylamide gels and transferred to polyvinylidene difluoride (PVDF) membranes (Merck Millipore, USA). The membranes were blocked with 5% nonfat dry milk in TBS-T (1% Tween 20 in Tris-buffered saline (TBS)) or Roti-Block (Carl Roth, Germany) and then incubated with the following primary antibodies diluted in TBS: mouse anti-GluA1-NT (Millipore, USA, MAB2263, 1:2000, #3847897), rabbit anti-PSD95 (Cell Signaling, USA, 2507, 1:2000, lot #4), Anti p44/42 MAPK (ERK1/2) antibody (Cell Signaling, USA, 9102, 1:10000, lot #30), Anti Phospho-p44/42 MAPK (Erk1/2) antibody (Cell Signaling, USA, 437, 1:10000, lot #21), Anti beta-Tubulin antibody (Sigma Aldrich, USA, T4026, 1:2.000, Lot# 0000324939), Anti AKT Antibody (Cell Signaling, USA, 9272, 1:1.000, Lot #30), Anti Phospho-AKT antibody (Cell Signaling, USA, 4060, 1:2000, Lot #27), Anti-GAPDH antibody (Cell Signaling, Cat 2118, 1:1000, Lot #16) and mouse anti-GAPDH (Abcam, United Kingdom, ab8245, 1:1000, lot #1035914-6). After three washes, the membranes were incubated with the following horseradish peroxidase-conjugated secondary antibodies diluted in TBS-T: sheep anti-mouse (GE Healthcare, USA, NA931, 1:20.000, lot #17556470 and #18111935) and donkey anti-rabbit (GE Healthcare, USA, NA9340, 1:25000) and Anti-rabbit antibody (SeraCare, USA, 5450-0010, 1:10.000, Lot #10571726) for 1 h at room temperature. In some experiments, the membrane was washed and stripped for further reincubation with another antibody. In that case, the membrane was incubated two times for 15 min in a glycin stripping solution (glycin 15 g, SDS 1 g, Tween20 10 mL, 800 mL dionized water), washed three times ten minutes in TBS-T, and then blocked and incubated following the regular procedure. The washed membranes were developed with a ChemiDoc MP imaging system (Bio-Rad) using an enhanced chemiluminescence detection kit (Thermo Fisher Scientific, USA). Band intensity was quantified by densitometry

with ImageJ 1.47 v software (National Institute of Health, USA) and normalized to the appropriate loading control.

## Real-time PCR
RNA was isolated from powdered frozen hippocampal samples using a NucleoSpin RNA kit (Macherey Nagel, Germany), and cDNA was prepared using Oligo d(T) primers and Ready-To-Go You-Prime First-Strand Beads (GE Healthcare, Germany). Real-time PCR was performed using a Takyon No Rox SYBR MasterMix dTTP Blue Kit (Eurogentec, Belgium) and a LightCycler 480 (Roche, Switzerland). The reference genes glyceraldehyde-3-phosphate dehydrogenase (GAPDH) and 40S ribosomal protein S18 (RPS18) were used as internal controls. The following primer pairs were chosen from previous literature[114] and were designed using PCR-Primer Design tool from Eurofins-Genomics. The following Primers were used (sequences provided as 5′-3′):

GluN2D (fwd: CTGTGTGGGTGATGATGTTCGT, rev: GTGAAGGTA GAGCCTCCGGG);

GAPDH (fwd: ACAACTTTGGTATCGTGGAAGG, rev: GCCATC ACGCCACAGTTTC); RPS18 (fwd: GCGGCGGAAAATAGCCTTTG, rev: GATCACACGTTCCACCTCATC).

Amplification was performed with an initial denaturation of 45 cycles of 95 °C for 10 s, followed by 45 cycles of 60 °C for 15 s and 72 °C for 15 s. A melting curve was obtained at the end of cycling to verify the amplification of a single PCR product. Relative gene expression of the Grin2d/GluN2D gene was calculated using the $2^{-\Delta Ct}$ method, where ΔCt represents the difference between the Ct value of the target gene and the geometric mean of the Ct values of the two reference genes[115].

## Behavioral procedures
Prior to all behavioral experiments, the mice were handled daily for 1 min over a time course of 5 days. Tunnel handling was used for all animal placement.

### Repeated stress model (RS)
Mice were forced to swim in a glass cylinder (Ø 26 cm, 60 cm high) filled to a depth of 25 cm with 25 °C water for 10 min on 5 consecutive days. The immobility time was measured in each session. After swimming, the animals were gently dried and kept in a cage under a heating light for an additional ten minutes. Afterward, the mice were allowed to rest for two days in their home cages. Mice were treated after the resting period, depending on the experimental group. For the measurement of escape behavior, an additional swim session was performed. For intraperitoneal injections, (±)-ketamine was diluted in NaCl (0.9%) and NAB-14 was dissolved in PEG 400 (40%), dimethylacetazolamide (10%), and 50% glucose (5%). Lorazepam was purchased as an injection solution (Tavor pro injectione®, 2 mg, Pfizer, Germany). A maximum volume of 0.2 ml was injected per mouse; the volume was individually adjusted to the weight of the animal to achieve the intended dose. As a control, the respective vehicle was administered.

### Behavioral readout
After the intervention, the animals were either subjected to an additional swim session (test day) to measure immobility time or subjected to the nosepoke sucrose preference test (NP-SPT) using the IntelliCage system (TSE Systems, Germany). For assessment of immobility time, the mice were videotaped, and two independent experienced raters who were blinded to the experimental groupings analyzed the videos. The mean values determined by both raters were used for further analysis. Immobility time was defined as the cumulative time that the animals remained stationary, during which only movements of the tail or forepaws necessary to keep the head above the water surface were made, the animals did not travel any distance and merely passively floated, no directed movement of the front paws was observed, and the body was mostly oriented parallel to the walls of the cylinder. A detailed description of this protocol and its analysis can be found here[57].

## Nose-poke sucrose preference test (SPT)

The IntelliCage system (TSE Systems, Germany), which allows automatic analysis of the spontaneous and exploratory behavior and drinking preference of rodents, was used. Radiofrequency identification transponders were implanted in the mice. IntelliCage comprises a common space in the middle and four measuring corners. Up to 16 group-housed mice can be given free access to food in the center, while water is supplied in the corners (by two zipper bottles each) behind electronically managed doors. The corners can be visited by only one mouse at a time. Using IntelliCage Plus software, the system records i) the number and duration of corner visits, ii) the number of nosepokes (approaches) on the doors, and iii) the number of licks on the drinking bottles. Initially, the mice were adapted to the IntelliCage for five days and provided free access to food and water in all corners. Then, for three days, the animals were habituated to sucrose (each corner contained one bottle with 1% sucrose solution and one with water; the contents of all bottles were exchanged every day) and subjected to the nosepoke protocol. The mice had to perform a nosepoke to open the doors and access both water and sucrose. Next, an NP-SPT paradigm[67] in which gradually increasing effort (number of nosepokes) was needed to access the sucrose-containing bottles for a short period (24 h) was used for the measurement of sucrose preference. In this protocol, each door opened in response to a nosepoke and closed after the animals drank from the sucrose-containing bottle for 5 s, while the mice were allowed to consume water from the water-filled bottles for an unlimited amount of time. The number of nosepokes needed to access the sucrose-containing bottle progressively increased (1, 2, 3, 4, 5, 6, 7) after each of the 10 licking sessions. This test was used to assess the drive for obtaining a reward; a decreased sucrose preference indicates reduced reward behavior. For every side, the number of licks was measured, and the mean sucrose preference was calculated as the percentage of the licks of the sucrose solution-containing bottles compared to the total number of licks. post 1 and post 2 represent two consecutive 24 h test periods.

## Open field test (OFT)

The test was performed in a square arena (50 × 50 cm) surrounded by a 35 cm high wall made of gray PVC. The mice were placed in the center of the field and allowed to move freely. The mice were video tracked for 10 min, and the time spent in the center (16% of the entire area of the box) was compared to the time spent outside the center (time in the center/time outside the center). Tracking and analysis were performed with EthoVision XT software (Noldus, Netherlands).

## Object location memory task (OLT)

For habituation, the mice were placed in the center of an empty square arena (50 × 50 cm) surrounded by a 35 cm high wall of the gray PVC and maintained there for 10 min. On the following day, the mice were placed in the empty arena for 10 min for a second habituation phase. Immediately thereafter, the mice were removed, and two slightly different objects were introduced into the arena (for location, see Fig. 7a). The animals were then placed back into the area and allowed to rest there for 10 min. Directly after the exploration of the two objects, the mice were injected (KET/NAB-14/saline) and placed back into their home cages. Six hours later, the mice were placed in the arena where the location of one of the two objects was changed (see Fig. 7a for the location of the object) and remained there for another 10 min. During all the time in the arena, the mice were video tracked using EthoVision XT software (Noldus, Netherlands).

## Rotarod

A ROTA ROD LE8205 (PanLab, Harvard Apparatus, Spain) was used for the experiments. This instrument automatically detects the latency to fall and enables simultaneous assessment of up to five animals. Before testing, the animals were exposed to three training sessions at 5 rpm for 1-3 min with a 10 min interval between each session. Animals that did not stay on the rod rotating at 5 rpm for at least 60 sec in the third training session were excluded from further testing. For testing, the animals were set on a rod at 4 rpm. The rod was set to accelerate from 4 to 40 rpm in 300 sec, and the latency to fall was measured. This procedure was repeated at 5, 20, 30 and 50 min postinjection, and mean values were calculated for each treatment condition.

## Locomotion

The test was performed in a square arena (50 × 50 cm) surrounded by a 35 cm high wall made of gray PVC. The mice were placed in the center of the field and allowed to move freely. Behavior was recorded for 10 min, and the total distance traveled was analyzed with EthoVision XT software (Noldus, Netherlands).

# Pharmacokinetic Study: Plasma concentrations and Brain Penetration of NAB-14

The pharmacokinetics and brain penetration of NAB-14 were investigated in Sprague Dawley male rats following single intravenous (i.v.) administration at 1 and 2 mg/kg.

NAB-14 (batch No.: 312155; purity: 98.26%) was given to 21 male rats/group (9-weeks old on the day of dosing) in *ad libitum* food regimen at 1 and 2 mg/kg by a single i.v. bolus injection.

NAB-14 was dissolved in 10% w/w NMP/ 10% w/w PEG400/ 5% w/w Solutol HS15/ 75% w/w sterile water for injection and administered at a dose volume of 5 ml/kg. The pH of the formulations was 6.59 and 6.41 at 0.2 and 0.4 mg/mL, respectively. The osmolality of the formulations was 1563 and 1569 mmol/kg at 0.2 and 0.4 mg/mL, respectively.

Composite plasma and brain profiles were obtained from three terminal animals at each time point (5 min, 15 min, 30 min, 1 h, 3 h, 6 h and 24 h). Pharmacokinetic parameters were evaluated after each administration. No deviations occurred during the study.

NAB-14 was extracted from rat plasma and rat brain homogenate by protein precipitation using acetonitrile containing Diclofenac as an internal standard. The rat brain homogenates were prepared from rat brain using phosphate buffer saline (PBS) as diluent in a ratio 1:4 (1 g of brain was homogenized with 4 mL of PBS) and were frozen at -80 °C. Extracts were analysed by LC-MS/MS using ZSprayTM interface with positive ion multiple reaction monitoring. These methods were qualified over the range 1 to 1000 ng/mL using a 10 µl aliquot of rat plasma and rat brain homogenate. Calibration standards were freshly prepared; quality control samples were prepared and stored at −80 °C together with the study samples until use. All analytical batches from which concentration results were reported met the acceptance criteria.

## Preparation of analytical solutions

Two set of analytical solutions (A/B) of NAB-14 from independent weighings were prepared and dissolved in the required volume of dimethylformamide (DMF) to give a 1 mg/ml stock solution of the free compound; one for calibration standards (A); and one for quality control samples (B), Table 1.

## Preparation of internal standard solutions

Diclofenac was dissolved in water/acetonitrile (50:50, v/v) to give a 1 mg/ml stock solution (C).

The stock and working solutions were stored at 4 °C, Table 2.

## Calibration standard and quality control preparation

Calibration standard samples and quality control samples were prepared on wet ice and thoroughly mixed as described below, 3–6.

## Extraction procedure

The samples were thawed on wet ice, and the tubes of study samples and/or CSs and/or QCs samples were vortexed. A 10 µl aliquot of

sample and/or CSs and/or QCs was placed into a tube and kept on wet ice. 100 µl of acetonitrile were added to the double blank(s). 100 µl of internal standard working solution (C1) was added to all other tubes. After capping the tubes, they were thoroughly vortexed. 100 µl of water were added to all tubes and again vortexed. The tubes were centrifuged for at least 10 minutes at approximately 3000 G and the supernatants were injected onto the LC-MS/MS system for analysis.

## Plasma concentrations of NAB-14 after oral application

The pharmacokinetics of NAB-14 was investigated in Sprague Dawley male rats following single oral (PO) administrations of two different test item formulations. NAB-14 (batch No.: 312155; purity: 98.26%, appearance: off white powder) was administered to 3 male rats/route/ formulation (9-weeks old on the day of first dosing occasion) in at

libitum food regimen. Animals were dosed at 10 mg/kg or 50 mg/kg by a single PO administration. NAB-14 was suspended in 20% w/v PEG400/ 5% w/v Transcutol HP/5% w/v Vitamin E TGPS in 200 mM Citrate Buffer for the first formulation (Formulation A) and in 25% w/v PEG400/5% w/ v Solutol HS-15/0.25% w/v MC 400 cp in 200 mM Citrate Buffer for the second formulation (Formulation B) and administered at a dose volume of 10 mL/kg. The pH of Formulation A was 3.53 and 3.58 at 10 and 50 mg/kg, respectively. The pH of Formulation B was 3.51 and 3.52 at 10 and 50 mg/kg, respectively. Individual serial plasma profiles were drawn from each animal. Pharmacokinetic parameters were evaluated after each administration. Body weights were within the normal range for animals of this strain and age. After treatment, NAB-14 was extracted from rat plasma, manually aliquoting samples kept on wet ice, using acetonitrile containing diclofenac as an internal standard. Extracts were analysed by LC-MS/MS using ZSprayTM interface with positive ion multiple reaction monitoring. This method (see section above) was qualified over the range 1 to 1000 ng/mL using a 10 µL aliquot of rat plasma. Calibration standards were freshly prepared; quality control samples were prepared and stored at -80 °C together with the study samples until use. All analytical batches met the acceptance criteria.

## Molecular modeling

The Schrödinger Software Suite 2021-4 (Schrödinger LLC, NY, USA) was used for modeling studies. The SMILES of NAB-14 was converted to a valid 3D Lewis structure using LigPrep (Schrödinger LLC, NY, USA) and was geometrically optimized with the OPLS4 force field[116]. The cryo-EM image of the heterotetrameric GluN1a/GluN2D NMDAR (PDB ID: 7YFF) structure was prepared for docking with the Protein PrepWizard[117]. This step involves adding missing hydrogen atoms and adjusting the ionization state of polar amino acids at neutral pH and, finally, energetically minimizing the structure using the OPLS4 force field. The geometric center of residue C590 (located on the GluN2D-

## Table 1 | Overview of analytical solutions for calibration and quality control in pharmacokinetic studies

| Working Solution | Final Concentration (µg/ml) | Vol of Spiking Solution | Vol of Water/Acetonitrile (50:50, v/v) (µl) |
|---|---|---|---|
| A1/B1 | 20 | 10 µL of solution A/B | 490 |

## Table 2 | Overview of solution for the diclofenac control in pharmacokinetic studies

| Working Solution | Final Concentration (ng/ml) | Vol of Spiking Solution | Volume of Acetonitrile (ml) |
|---|---|---|---|
| C1 | 250 | 125 µl of stock solution C | 500 |

## Table 3 | Overview of calibration standard samples for plasma homogenates in pharmacokinetic studies

| Calibration Standard Concentration (ng/ml) | Volume of Spiking Solution (µl) | | | | | | | Volume of Rat Plasma Homogenate (µl) |
|---|---|---|---|---|---|---|---|---|
| | CS3 10 ng/ml | CS4 40 ng/ml | CS5 100 ng/ml | CS6 400 ng/ml | CS7 800 ng/ml | CS8 1000 ng/ml | A1 20 µg/ml | |
| 1000 (CS8) | - | - | - | - | - | - | 25 | 475 |
| 800 (CS7) | - | - | - | - | - | - | 20 | 480 |
| 400 (CS6) | - | - | - | - | - | 200 | - | 300 |
| 100 (CS5) | - | - | - | - | 75 | - | - | 525 |
| 40 (CS4) | - | - | - | 50 | - | - | - | 450 |
| 10 (CS3) | - | - | 50 | - | - | - | - | 450 |
| 2 (CS2) | - | 25 | - | - | - | - | - | 475 |
| 1 (CS1) | 50 | - | - | - | - | - | - | 450 |

## Table 4 | Overview of calibration standard samples for brain homogenates in pharmacokinetic studies

| Calibration Standard Concentration (ng/ml) | Volume of Spiking Solution (µl) | | | | | | | Volume of Brain Homogenate (µl) |
|---|---|---|---|---|---|---|---|---|
| | CS3 10 ng/ml | CS4 40 ng/ml | CS5 100 ng/ml | CS6 400 ng/ml | CS7 800 ng/ml | CS8 1000 ng/ml | A1 20 µg/ml | |
| 1000 (CS8) | - | - | - | - | - | - | 20 | 380 |
| 800 (CS7) | - | - | - | - | - | - | 16 | 384 |
| 400 (CS6) | - | - | - | - | - | 100 | - | 300 |
| 100 (CS5) | - | - | - | - | 25 | - | - | 175 |
| 40 (CS4) | - | - | - | 25 | - | - | - | 225 |
| 10 (CS3) | - | - | 25 | - | - | - | - | 225 |
| 2 (CS2) | - | 10 | - | - | - | - | - | 190 |
| 1 (CS1) | 20 | - | - | - | - | - | - | 180 |

TMD M1 helix) was considered the grid centroid, and flexible docking was carried out using Glide with the SP scoring function[118].

## Quantification and statistical analysis

All given values are the mean ± SEM, and the error bars represent the SEMs in the figures; n represents the number of experiments. For the statistical analyses, GraphPad Prism version 8.3.0 (GraphPad Software, USA) was used. To test for significance within experimental groups (e.g., baseline vs. effect), we used a paired t test or a Wilcoxon signed rank test. To compare two experimental groups, either an unpaired t test or the Mann-Whitney test was performed. To compare several time points in one experimental series, repeated-measures ANOVA followed by Dunnett's correction was used. To test for differences among several treatment groups, ordinary one-way ANOVA followed by Dunnett correction was used. For experiments shown in Fig. 1, in which the influence of different concentrations of a compound on NMDAR currents was compared between interneurons and pyramidal cells, a two-way ANOVA was performed to assess main effects of cell type and concentration, as well as their interaction. To test for differences between groups with two variables, repeated-measures ANOVA within factors was conducted, and for estimation of effect sizes, ETA squared values were calculated. In cases of sphericity violation, the Greenhouse-Geisser adjustment was applied. Outliers were excluded using ROUT methods set at 1%. All the statistical tests were two-tailed, if applicable. Significance levels are depicted by asterisks in the figures: $*p < 0.05$, $**p < 0.01$, $***p < 0.001$, $****p < 0.001$.

## Sex aspects

All experiments were performed with an equal distribution of male and female mice. For all experiments, a separate exploratory analysis focusing on the potential effects of sex on the experiments was performed, but no relevant differences were observed. However, our experiments were not adequately powered to reliably detect sex differences.

## Analysis of patch-clamp whole-cell data

**Rs and Rm.** Whole-cell currents in response to 50 ms/-5 mV hyperpolarizing pulses were analyzed. Rs was calculated by dividing -5 mV by the maximal current amplitude (Is). Rs represents the electrical resistance between the pipette and cell soma in whole-cell measurements.

Rm was calculated by dividing -5 mV by the stable current amplitude (Im) after the initial current peak and represents the input resistance of the patched cell.

**Analysis of LTP.** In the LTP figures, each dot represents six consecutive averaged maximum EPSP amplitudes ± SEM. The horizontal bars at 25–30 min indicate the significant differences between baseline and post-induction EPSP amplitudes within each experimental group. The mean amplitude of 30 consecutive EPSPs was calculated before the induction protocol to obtain the baseline amplitude. Twenty-five minutes after LTP induction, the mean amplitude of 30 consecutive EPSPs was calculated and compared to the baseline value from the same experiment with a two-tailed t test. Changes in EPSP amplitudes are expressed as percentages of the baseline measurements and were analyzed for each experimental group to quantify the degree of LTP.

**Paired-pulse ratio (PPR).** In most experiments, every 10th EPSP was replaced by two consecutive EPSPs (EPSP 1/2) with an interval of 50 ms, and their maximal amplitudes were measured. The PPR was then calculated as

$$PPR = (EPSP\_2(mV))/(EPSP\_1(mV)),$$ and the mean PPR values were compared between baseline values before induction of the weak aLTP and those 25-30 min after induction. A change in the PPR commonly indicates a change in the probability of presynaptic transmitter release, whereas no change indicates a postsynaptic mechanism[119–121].

**MICA-EPSP amplitude.** The maximum EPSP amplitudes from min 0-10 were averaged and compared to the average maximum EPSP amplitudes from min 20-30.

**MICA-AP-conversion.** A 10-minute baseline was recorded during continuous MICA protocol stimulation at 0.2 Hz with EPSPs adjusted close to AP-threshold levels (ca. 15-25 mV). Thereafter, test substances were added to the bath solution, and recording continued for 20 minutes. AP spikes were counted and calculated as the mean number of APs over periods of 10 minutes.

**MICA-Decay Tau.** The trace of each EPSP was fitted using the following formula:

$$y(x) = RMP + Amp_1 * \left\{ 1 - \exp\left( -\frac{x - t0_1}{Tau_{on}} \right) \right\} * \exp\left( -\frac{x - t0_1}{Tau_{decay}} \right)$$

where RMP is the resting membrane potential (mV) and AMP_1 (mV) is the maximum EPSP amplitude. Tau_on (ms) is the onset tau of the EPSP, and Tau decay (ms) is the decay. For each cell, the mean EPSP-decay tau time constants were calculated and averaged for each experimental group. Fitmaster NEXT (HEKA, Germany) was used for fitting.

**Fitting (Figure 4).** For the NMDAR current experiments and weak-aLTP experiments with NAB-14, a nonlinear regression was fitted to the

**Table 5 | QC Concentration for standard samples for plasma homogenates in pharmacokinetic studies**

| QC Concentration(ng/ml) | Volume of Spiking Solution (µl) | | | | Vol of Rat Plasma (µl) |
|---|---|---|---|---|---|
| | M QC 300 ng/ml | H QC 750 ng/ml | B1 20 µg/ml | B 1 mg/ml | |
| 37500 (DIL QC)[1] | - | - | - | 22,5 | 577,5 |
| 750 (H QC) | - | - | 15 | - | 385 |
| 300 (M QC) | - | 200 | - | - | 300 |
| 3 (L QC) | 10 | - | - | - | 990 |

**Table 6 | QC Concentration for standard samples for brain homogenates in pharmacokinetic studies**

| QC Concentration(ng/ml) | Volume of Spiking Solution (µl) | | | | Vol of Brain Homogenate (µl) |
|---|---|---|---|---|---|
| | M QC 300 ng/ml | H QC 750 ng/ml | B1 20 µg/ml | B 1 mg/ml | |
| 37500 (DIL QC)[a] | - | - | - | 7,5 | 1925 |
| 750 (H QC) | - | - | 15 | - | 385 |
| 300 (M QC) | - | 100 | - | - | 150 |
| 3 (L QC) | 10 | - | - | - | 990 |

[a]Prepared for qualification only and diluted 50-fold (10 µl diluted with 490 µl of Rat brain homogenate).

measured x/y values, and the IC50 values were calculated (x = NMDAR current/amount of LTP, y = KET/NAB-14 concentration). The model used was as follows:

$$IC_{50} = \frac{ICF}{\left(\frac{F}{100-F}\right)^{\frac{1}{Hillslope}}}$$

$$Y = Bottom + \frac{Top - Bottom}{1 + \left(\frac{IC50}{X}\right)^{HillSlope}}$$

(F = 50)

For weak LTP experiments with KET, a Gaussian distribution model was fitted to the measured x/y values (x = amount of LTP, Y = KET concentration). The model used was as follows:

$$Y = Amplitude * \exp\left(-0.5 * \left(\frac{X - Mean}{SD}\right)^2\right)$$ *Analysis of Maximal Drug Effects in weak-LTP experiments EPSCs.*

To determine the concentration at which each drug exerted its maximal effect, we extracted the peak of the fitted dose–response curves. For both KET and NAB14, we used the Excel function =MAX(B:B) to identify the highest value of the fitted curve, where column B contained the fitted values. The corresponding drug concentration was then obtained using =INDEX(A:A, MATCH(C2, B1:B1000, 0)), with column A listing the tested concentrations and C2 containing the maximum value.

**Analysis of Maximal Drug Effects in Cell-Type Differences of NMAR EPSCs.** To compare NMDAR current inhibition between interneurons and pyramidal cells, we first subtracted the fitted values of the interneuron curve (which showed stronger inhibition) from the corresponding fitted values of the pyramidal cell curve. This subtraction was performed point-by-point across all concentrations. The concentration at which this difference was maximal was determined using the same =MAX() and =INDEX()/=MATCH() functions as described above. This analysis was carried out separately for each drug.

**CV analysis.** The slope of the EPSP rise at 20% to 40% of its maximal amplitude, at which point an approximately linear increase in its voltage could be assumed, was fitted with Fitmaster software (HEKA, Germany). We analyzed 20 EPSPs before and 20 EPSPs 25-30 min after NAB-14 wash-in in the MICA protocol for each experiment. The coefficient of variation (CV) is the standard deviation of the EPSP slopes divided by the mean. The inverse square of the CV of the postwash-in slopes was divided by the inverse square of the prewash-in slopes and plotted against the corresponding normalized slopes. Pre- and postwash-in measurements from a single experiment are connected by a line. In a standard quantal model for synaptic transmission, synaptic responses are affected either by presynaptic changes in the number of release sites and/or the probability of release or by modifications of the postsynaptic response to a single vehicle. A change in the ratio of $CV^{-2}$ reflects a presynaptic action, whereas horizontal lines in the $CV^{-2}$ plot indicate a change in postsynaptic responsiveness[120,122,123].

## Morphological analysis

All quantifications were performed by a blinded researcher. For GAD65 quantification, all images were normalized using the quantile-based normalization plugin Fiji (https://www.longair.net/edinburgh/imagej/quantile-normalization/). The intensity distribution of the images was normalized using 256 quantiles for each staining. The synaptic boutons were extracted from the background using the 3D Weka Segmentation plugin (https://imagej.net/Trainable_Weka_Segmentation) after manual selection of signal and background samples for training. The Fiji built-in plugin 3D object counter was then used to count and measure every object (cluster of GAD65-positive signal).

For VGLUT1 quantification, the raw images were first deconvoluted using Huygens software (Scientific Volume Imaging, The Netherlands). VGLUT1 density and volume were then quantified using built-in Spot detection analysis with Imaris 10.0 software (Oxford Instruments, United Kingdom).

### Reporting summary
Further information on research design is available in the Nature Portfolio Reporting Summary linked to this article.

## Data availability
The electrophysiological, imaging, pharmacokinetic, and behavioral data generated in this study have been deposited in the Figshare database under accession code https://doi.org/10.6084/m9.figshare.30436471. The Supplementary data are available in the Figshare database under accession code https://doi.org/10.6084/m9.figshare.30436492. Source data is available in Source data table. All materials are commercially available. Source data are provided with this paper.

## Code availability
All code is available including a how to use readme description under: https://doi.org/10.6084/m9.figshare.29245961.

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

## Acknowledgements

We thank Daniel Zell, Magdalena Weidner-Büchele, and Johanna Diemer for preparatory experiments, the Analytical and High-Resolution Microscopy in Biomedicine Platform of SGIker (EHU/ERDF, EU, Bilbao) for technical and human support. Funding: Berta-Ottenstein-Program for Clinician Scientists, Faculty of Medicine, University of Freiburg, Germany, personal scholarship (S.V.). Medical Research Foundation, France (FRM) grant AJE201912009450 (T.S.). University of Strasbourg Institute of Advanced Study (USIAS) grant 2020-035 (T.S.). The German Research Foundation (DFG) grants SE2666/2-1 (T.S., C.N.) and NO370/7-1 (C.N.). Centre National de la Recherche Scientifique, France, grant CNRS UPR3212 (T.S.). GO-Bio-initial grants INNOVADE 16LW0223 and 16LW0455 (C.N.) were obtained from the German Federal Ministry for Education and Research (BMBF). Images were created with Biorender.com, Vestring, S. (2025) https://BioRender.com/dpm4zwz.

## Author contributions

Conceptualization, S.Ve., N.G., M.V., T.S., S.G., J.Bi., C.N.; Methodology, S.Ve., M.V., N.G., D.S., T.S., F.H., S.G., S.K., M.W., P.P., C.N.; Investigation, S.Ve., M.V., M.C.P., M.E., A.L., J.L., L.M.W., F.E., J.E., J.M., D.W., A.T., E.W., L.M.B., S.Vo., F.H., J.B., F.H., C.V., D.J., S.K., M.V., J.P.G., J.Wa., J.We., P.L., S.B., L.S. S.Z., P.P., D.S. G.S., J.Br., G.L., C.D.V., E.G., N.G., T.S.; Formal analysis, S.Ve., M.C.P., L.E.S., P.P., M.V., S.G., A.M., N.G., T.S., C.N.; Visualization, S.Ve., A.L., A.M., S.K., L.E.S., M.W., S.G., C.N.; Funding acquisition, S.Ve., K.D., T. S., C.N.; Project administration, S. Ve., T.S., M.V., C.N.; Supervision, S.Ve., N.G., K.D., S.G., S.K., T.S., M.V., C.N.; Writing – original draft, S.Ve., C.N.; Writing – review & editing, S.Ve., M.C.P., K.D., T.S., J.Bi., C.N.

## Funding

## Competing interests

C.N., S.Ve., and K.D. received lecture fees and advisory board honoraria from Johnson & Johnson, the manufacturer of Esketamine. C.N. received research support as a principal investigator in clinical trials sponsored by Johnson & Johnson. C.N. received honoraria as a member of a DMC board by Novartis. C.N. and S.Ve. are named coinventors on a patent filed by the University of Freiburg for the use of GluN2D inhibitors in the treatment of depression (EP 22153076.9/ WO 2023/144163). T.S. received honoraria consulting Primetime Life Sciences, LLC. All other authors declare no competing interests.

## Additional information

[1]Department of Psychiatry and Psychotherapy, Medical Center – University of Freiburg, Faculty of Medicine, University of Freiburg, Freiburg, Germany. [2]Berta-Ottenstein-Program for Clinician Scientists, Faculty of Medicine, University of Freiburg, Freiburg, Germany. [3]Department of Medicine I, Medical Center - University of Freiburg, Faculty of Medicine, University of Freiburg, Freiburg, Germany. [4]Centre National de la Recherche Scientifique (CNRS) UPR3212, Université de Strasbourg, Institut des Neurosciences Cellulaires et Intégratives (INCI), Strasbourg, France. [5]University of Strasbourg, Institute for Advanced Study (USIAS), Strasbourg, France. [6]Core Facility Signaling Factory, Centre for Biological Signaling Studies (BIOSS), University of Freiburg, Freiburg, Germany. [7]Department of Cell Biology and Histology, Faculty of Medicine and Nursing, University of the Basque Country (EHU), Leioa, Spain. [8]Department of Physiology and Cell Biology, and the School of Brain Sciences and Cognition, Ben-Gurion University of the Negev, Beer-Sheva, Israel. [9]Ikerbasque, Basque Foundation for Science, Bilbao 48013, Spain. [10]Institute of Pharmaceutical Sciences, University of Freiburg, Freiburg, Germany. [11]German Center for Mental Health (DZPG), Partner Site Berlin/Potsdam, Berlin, Germany. [12]Department of Biomedicine, University of Basel, Basel, Switzerland. ✉e-mail: claus.normann@uniklinik-freiburg.de

