## [Transparent Peer Review file · Nature Communications]

The NMDA receptor subunit GluN2D is a potential target for rapid antidepressant action

Corresponding Author: Professor Claus Normann

Version 0:

Reviewer comments:

Reviewer #1

(Remarks to the Author)

This is an excellent paper thoroughly dissecting the role of the GluN2D receptor subtype in the antidepressant response to ketamine. The authors use a very wide range of approaches ranging from molecular docking to electrophysiology to translationally relevant behaviors in chronically stressed mice. An additional strength of the paper is that although they primarily rely on a selective pharmacological antagonist, they also use an siRNA approach to replicate key pharmacological results – an approach rarely taken in similar studies. The paper is clearly written, the data are clearly presented and tested statistically, and the conclusions are justified. I have only the following relatively minor comments.

1. Intro: The work of Hodayoun and Moghaddam, 2007 (PMID: 17959792) should be cited around lines 61-64 as evidence that ketamine does in fact produce selective disinhibition. Also, around lines 65-66, the authors should cite evidence that ketamine promotes EEG gamma oscillations, as this is the likely explanation for the increase in glutamate release and perhaps ketamine's therapeutic actions.
2. Line 105: Correct reference for first mention of NAB-14 should be #26 (Swanger et al., 2018). Ref 18 does not mention it.
3. Fig 1 d,e: As written, the two separate rmANOVAs demonstrate that ketamine has significant effects in PCs and INTs. The authors have not tested whether the effects in INTs are greater than in PCs, which is the conclusion they draw in the text.
4. Fig 2b: I am confused why the 10uM dose of NAB did not affect fEPSPs, since fig 1d demonstrates that they are equally effective on INT NMDARs. I am also confused about the differences in the two graphs. The graph at left looks like a 3-fold increase in EPSP amplitude for 20uM NAB, whereas the graph at right shows doubling. Similarly 10uM has no effect at left, but produces a 50% increase at right.
5. In the discussion of the results of Fig 3c-d, the authors should reference the results of Zanos et al. 2023 (PMID: 36596696) demonstrating similar U-shaped dose response relationships in antidepressant-relevant behaviors.
6. This is an incredibly thorough paper with extensive data. I am reluctant to ask for more data on top of the plentiful work they provide, however, I suggest to the authors that they consider examining the role of NR2D in two aspects of ketamine's known behavioral side effect profile: conditioned place preference (as indicator of abuse liability) and prepulse inhibition (an indicator of potential risk for psychosis-like responses). This is by no means a demand! It is likely, but by no means certain, that NR2D inhibition would be involved. If they were not, the authors would strongly amplify the suggestion that selective NR2D NAMs have novel therapeutic value.

Reviewer #2

(Remarks to the Author)

This well prepared manuscript described a role of GluN2D NMDA receptor subunit in mediating rapid antidepressant actions. Using ephys and behavioral analysis, the authors demonstrate that inhibition of GluN2D in adult hippocampus can reduce GABAergic interneurons controlled feedback and feed-forward inhibitory circuits, resulting increased hippocampal network activity and synaptic plasticity. Additionally, in mouse models of depression, GluN2D inhibition recovered synaptic excitation-inhibition balance, reversed long-term potentiation deficits, and mimics the cellular and behavioral antidepressant-like actions of ketamine with fewer side effects. Together, this work provides new insights into mechanisms of fast-acting antidepressant actions and suggests GluN2D as a novel potential drug target. The experiments are carefully designed and executed.

Major points:

- 1) Rodent models just mimic certain aspects of depression at most. Usually conclusions from animal models are summarized as depression-like, or anti-depressant-like. Also before the drug targets confirmed in clinical trials, they are usually termed as potential drug targets.
- 2) 10 and 20uM NAB-14 inhibits NMDAR-current from both hippocampal interneuron and pyramidal neurons very similarly (Fig. 1d), why do they have very different effects on the EPSP also measured from hippocampus (Fig.2b)? Could off-target effects other than GluN2D contribute to this?
- 3) NAB-14 and ketamine treatments led to quite different impacts on vGlu1 and GAD65 staining in hippocampus. Is it possible these reflect differences in underlying mechanisms between NAB-14 and ketamine? Since ketamine also led to different impacts as NAB-14 in other experiments, such as Fig.2b-c, it seems like an overstatement to claim that "GluN2D antagonism could fully mimic the cellular and behavioral antidepressant actions of ketamine".
- 4) What is the knockdown efficiency in Fig. 4d in the targeted brain regions? The western blot sample images in sFig.5b is very difficult to understand. There are barely any bands. How specific is this antibody?
- 5) NAB-14 treatment boosts both excitatory and inhibitory (Fig. 5c, measured with GAD65) synaptic markers. Why does the manuscript conclude that "GluN2D inhibition shifts synapses toward excitation (Fig. 5 title)?"
- 6) Which brain region is responsible for GluN2D inhibition-induced antidepressant-like effects?
- 7) The drug concentration and doses are both described using numbers without units throughout the figures. This is confusing.
- 8) Does the pharmacokinetics/pharmacodynamics property of NAB-14 support that inhibition of GluN2D is responsible for the behavioral effects that NAB-14 treatment provides?
- 9) Is GluN2D required for ketamine's antidepressant like effects?
- 10) Recent publications suggest that different GluN subtypes lead antidepressant-like effects through different mechanisms. The relationship between GluN2D and other GluN subunits associated antidepressant-like mechanisms should be discussed and better explored experimentally, especially with GluN2B, which also works through a mechanism from inhibitory neurons.
- 11) Why aLTP can be induced (Supplementary Figure 3d), while weak-aLPT is inhibited (Fig 3c) by 10uM ket.
- 12) Does inhibition of GluN2D change baseline excitatory or inhibitory synaptic transmission?
- 13) Why compare immobile time of the same CDM mice on day 5 with it on day 8 (Fig 6a), instead of comparing the CDM-Sal mice to drug-treated on day 8?
- 14) Is SPT initiated 24 hours after siRNA injection in Fig 6b? How fast can siRNA knockdown GluN2D in vivo?
- 15) There is previous literature showed that S-ketamine at dose of 10mg/kg significantly increased travel distance (PMID:26327690), while Fig 7c showed the locomotion of mice was not increased after ket treatment. What makes the difference?
- 16) Is the GluN2D expression changed in CDM mice of both sex? Also, it will be better to show the data from different genders in different colors.

Minor points:

- 1) The mouse inbred line C57BL/6J is widely used. And there were behavioral differences between C57BL/6J and C57BL/6N mice (PMID: 23902802). Why did the authors choose C57BL/6N mice as experimental subjects?

Reviewer #3

(Remarks to the Author)

The manuscript by Vestring et al. investigates the role of the GluN2D subunit of NMDA receptors in producing rapid antidepressant effects in mice. By comparing the effects of a GluN2D antagonist NAB-14 with ketamine, the author demonstrated that inhibition of GluN2D reduced NMDAR currents in interneurons but did not affect pyramidal neurons within hippocampal slices. Consistently, inhibiting GluN2D or somatostatin (SOM)-positive interneuron activity facilitated long-term potentiation and increased synaptic protein expression in the hippocampus. Further investigation into the role of GluN2D antagonism using a mouse stress model revealed that GluN2D inhibition restored both synaptic and behavioral deficits induced by repeated stress exposure. Overall, this manuscript is well-written, and the findings are compelling. However, there are concerns about the methods used for GluN2D knockdown in the stress model and a lack of rigor in morphological analysis. I hope the authors will consider addressing the suggestions below to enhance the manuscript further.

1. While the authors' decision to conduct GluN2D knockdown in the nervous system to examine its role in stress-induced behaviors is commendable, given the electrophysiological data from hippocampal slices, it would be highly beneficial to investigate the specific effects of GluN2D knockdown on SOM interneurons in the hippocampus. This would offer a more comprehensive understanding of the research findings.

2. The authors claim that GluN2D inhibition reverses stress-induced impairment in synaptic morphology and density. However, they have only conducted immunostaining and western blotting of synaptic proteins without directly analyzing synaptic morphology. It would be valuable to assess the density and morphology of dendritic spines to provide clearer insights.

3. What is the duration of the effect of NAB-14? A notable feature of ketamine's antidepressant action is not only its rapid

onset but also its prolonged effects. A comparative analysis would be highly informative.

4. How does stress activate GluN2D? Specifically, what mechanisms might underlie the selective recruitment of GluN2D in response to stress?

5. In Fig. 5e and 5f, the values are normalized to those of CDM. Why were actual values not used for comparison? Is this normalization intended to account for experimental variations? It would be beneficial to normalize against the values of naïve mice to illustrate the variability observed under CDM.

6. Regarding Fig. 7c and 7d, how much time elapsed between drug administration and the OFT? It is known that a single ketamine administration produces anxiolytic effects 24 hours post-administration.

7. The western blot signals in Supplementary Fig. 5b could be clearer. The author should present the entire membrane or indicate molecular size alongside the membranes.

8. Could the authors specify the post-hoc tests used in this study not only in the Supplementary Table but also in the Figure legends?

9. The layout of Supplementary Table 1 in the PDF appears distorted.

10. Fig .6a is not cited anywhere in the manuscript. Please include a citation for this figure.

Version 1:

Reviewer comments:

Reviewer #2

(Remarks to the Author)

The authors have provided adequate explanations in the rebuttal letter or changes in the manuscript to address most of my concerns.

(Remarks on code availability)

Reviewer #3

(Remarks to the Author)

The authors appear to have appropriately addressed most of the concerns raised by this reviewer. However, the reviewer would like to seek clarification regarding the authors' interpretation of the neuronal morphological analysis presented in the newly added Fig. 6a and 6b. While GluN2D inhibition resulted in an increase in spine density, consistent with the authors' overall claims, repeated stress did not appear to reduce spine density. This finding seems to contrast with the observed decrease in synaptic protein expression and spontaneous EPSC frequency following repeated stress. The reviewer would appreciate it if the authors could elaborate on this apparent discrepancy, for instance, whether the intensity or duration of the stress protocol employed in this study may have been insufficient to elicit detectable changes in spine density.

(Remarks on code availability)

Point-to-Point-Response to the Reviewers

Following the recommendations of the journal, we have uploaded all raw data, statistical analyses and fiber photometry recordings on figshare.com. These are the private links for the reviewing process:

<https://figshare.com/s/a4282f8880d4b1fb7264> (Main figures)

<https://figshare.com/s/e68891c87d7f616787e7> (Supplementary figures)

<https://figshare.com/s/704513871cbb5e170dc6> (fiber photometry recordings)

The doi-links in the manuscript will be activated after publication.

Reviewer #1:

This is an excellent paper thoroughly dissecting the role of the GluN2D receptor subtype in the antidepressant response to ketamine. The authors use a very wide range of approaches ranging from molecular docking to electrophysiology to translationally relevant behaviors in chronically stressed mice. An additional strength of the paper is that although they primarily rely on a selective pharmacological antagonist, they also use an siRNA approach to replicate key pharmacological results – an approach rarely taken in similar studies. The paper is clearly written, the data are clearly presented and tested statistically, and the conclusions are justified. I have only the following relatively minor comments.

We would like to express our gratitude to the reviewer for the overall positive feedback and constructive recommendations. We highly appreciate the thoughtful and thorough comments. Based on the reviewer's comments, we have performed a number of additional experimental series, and think that we were able to improve the clarity and strength of the manuscript. Thank you for your valuable input.

1. Intro: The work of Homayoun and Moghaddam, 2007 (PMID: 17959792) should be cited around lines 61-64 as evidence that ketamine does in fact produce selective disinhibition. Also, around lines 65-66, the authors should cite evidence that ketamine promotes EEG gamma oscillations, as this is the likely explanation for the increase in glutamate release and perhaps ketamine's therapeutic actions.

We have added several important citations on the preferential disinhibition of interneurons even by non-selective NMDAR antagonists and on the increase of glutamate levels due to a ketamine-induced increase in gamma oscillations (lines 70-75).

2. Line 105: Correct reference for first mention of NAB-14 should be #26 (Swanger et al., 2018). Ref 18 does not mention it.

We apologize for the unclear referencing. We have now added the Swanger citation at the first mention of NAB-14 in the Results section (line 115) and relocated the Perszyk reference to a more

contextually appropriate position (line 116), where the low GluN2C expression in adult mice is specifically addressed.

3. Fig 1 d,e: As written, the two separate rmANOVAs demonstrate that ketamine has significant effects in PCs and INTs. The authors have not tested whether the effects in INTs are greater than in PCs, which is the conclusion they draw in the text.

We thank the reviewer for this important and well-reasoned observation. We agree that our initial analysis and wording did not adequately support a direct statistical comparison of effect magnitude between PCs and INs, and we appreciate the opportunity to clarify this point.

To address the issue, we reanalyzed the ketamine dataset using a mixed-effects model with cell type and concentration as fixed factors (Fig. 1e). This analysis revealed significant main effects of both concentration ($p < 0.0001$) and cell type ($p = 0.0042$). Thus, while ketamine suppressed currents in both cell types in a concentration-dependent manner, the overall current amplitude was significantly lower in INs across conditions. Importantly, post hoc comparisons at individual concentrations confirmed that INs were significantly more affected than PCs, providing robust statistical support for a stronger effect in this cell type. In Fig. S1f–g, we illustrate the dose-dependent suppression of currents in both PCs and INs.

For NAB-14, we applied the same mixed-effects model (Fig. 1d). In the original dataset, we observed a significant main effect of cell type and a borderline significant effect of concentration. Given the limited sample size, we conducted an independent replication under identical experimental conditions to strengthen the conclusions from this critical experiment. After confirming consistency between the original and replication datasets, we merged them and repeated the analysis. The combined data revealed robust and consistent main effects of both concentration and cell type, with significantly greater current suppression in INs at multiple concentrations. In Fig. S1d–e, we show that NAB-14 produces a clear dose-dependent inhibitory effect in INs, but not in PCs. Notably, increasing the sample size did not alter the overall results. We have updated the figure legends, statistical reporting, and text throughout the manuscript to reflect these revised and more rigorous analyses.

4. Fig 2b: I am confused why the 10uM dose of NAB did not affect ffEPSPs, since fig 1d demonstrates that they are equally effective an INT NMDARs. I am also confused about the differences in the two graphs. The graph at left looks like a 3-fold increase in EPSP amplitude for 20uM NAB, whereas the graph at left shows in doubling. Similarly 10uM has no effect at left, but produces a 50% increase at right.

We thank the reviewer for this important and carefully considered comment, which raises two distinct issues.

First, regarding the discrepancy between the left and right panels of Fig. 2b:

We identified an error in the original version of the left panel, which included in the 20 μ M NAB-14 group one outlier and one recording in which only a few EPSPs were detected - due to the strong effect of NAB-14, some responses had converted into action potentials. These recordings have now been excluded from the analysis, and the figure has been corrected accordingly. We apologize for this oversight and any resulting confusion.

We also acknowledge that the visual presentation in the previous version was suboptimal. Specifically, all baseline measurements were plotted using identical symbols, which obscured group-specific differences. Notably, the control group exhibited relatively high baseline EPSP

amplitudes, whereas the 10 μ M NAB-14 group started with lower values, contributing to a potentially misleading visual impression. We have revised the plot to assign distinct symbols to each experimental group, thereby enhancing clarity and interpretability.

Second, regarding the apparent discrepancy between the effects of 10 μ M NAB-14 on NMDAR currents in interneurons (Fig. 1d) and on ffEPSPs (Fig. 2b):

This is indeed an important point, and we appreciate the opportunity to clarify it. Due to word count constraints, we did not elaborate on this interpretation in the original manuscript - an omission we now recognize as a mistake. We have revised the text to explicitly address this issue and clarify our reasoning.

In the ffEPSP recordings, we observed a trend toward increased EPSP amplitudes with 10 μ M NAB-14, although this effect did not reach statistical significance. By contrast, the same concentration produced a significant enhancement of NMDAR-mediated currents in feedback interneurons (Fig. 1d). While this may appear contradictory at first glance, it likely reflects differences in both cell-type-specific receptor expression and experimental stimulation paradigms.

Most importantly, the two experiments target distinct interneuron populations. NMDAR currents in Fig. 1d were recorded from feedback interneurons - primarily SOM-positive cells in stratum oriens - known to express high levels of the GluN2D subunit, the selective target of NAB-14. In contrast, the ffEPSPs in Fig. 2b likely reflect activation of feedforward interneurons (predominantly PV-positive basket cells, though other subtypes may contribute), which are recruited by Schaffer collateral stimulation and express lower levels of GluN2D. These differences in receptor distribution likely account for the stronger NAB-14 sensitivity observed in feedback interneurons.

In addition, the stimulation paradigms differed substantially. Recordings from feedback interneurons involved direct and strong stimulation in stratum oriens, likely producing near-maximal NMDAR activation, which enhances the detectability of even modest pharmacological effects. By contrast, the ffEPSP recordings relied on subthreshold stimulation of Schaffer collaterals, leading to the activation of fewer receptors. This reduced level of activation may obscure more subtle modulatory effects, particularly at lower NAB-14 concentrations.

We have now incorporated this explanation into the revised manuscript (lines 163-167) to facilitate the reader's understanding of the observed differences between the two experimental conditions.

5. In the discussion of the results of Fig 3c-d, the authors should reference the results of Zanos et al. 2023 (PMID: 36596696) demonstrating similar U-shaped dose response relationships in antidepressant-relevant behaviors.

We thank the reviewer for highlighting these striking similarities. The reference has been added in line 237.

6. This is an incredibly thorough paper with extensive data. I am reluctant to ask for more data on top of the plentiful work they provide, however, I suggest to the authors that they consider examining the role of NR2D in two aspects of ketamine's known behavioral side effect profile: conditioned place preference (as indicator of abuse liability) and prepulse inhibition (an indicator of potential risk for psychosis-like responses). This is by no means a demand! It is likely, but by no means certain, that NR2D inhibition would be involved. If they were not, the authors would strongly amplify the suggestion that selective NR2D NAMs have novel therapeutic value.

We fully agree that investigating the behavioral profile of GluN2D-selective NAMs - particularly with respect to abuse liability (e.g., conditioned place preference) and psychosis-like effects (e.g., prepulse inhibition) - would provide valuable insights into the therapeutic window and safety profile of this compound class. These aspects are a key focus of our planned follow-up studies.

In addition to the points raised by the reviewer, we are particularly interested in assessing the effects of NAB-14 on cognitive domains such as working memory, attention, and cognitive flexibility, as well as on social behaviors including social interaction and motivation. We also plan to investigate motivational aspects of exploratory behavior, which are relevant to both negative symptoms and general affective state.

However, we respectfully note that the current study was designed to elucidate the cellular and circuit-level mechanisms of GluN2D inhibition and to evaluate its antidepressant efficacy in relevant preclinical models. A comprehensive behavioral characterization lies beyond the scope of this mechanistic manuscript.

We greatly appreciate the reviewer's thoughtful suggestion and have added a short statement at the end of the Discussion section (lines 525-527) to highlight these important future directions, which we agree are essential for fully assessing the translational potential of GluN2D-selective NAMs.

Reviewer #2:

This well prepared manuscript described a role of GluN2D NMDA receptor subunit in mediating rapid antidepressant actions. Using ephys and behavioral analysis, the authors demonstrate that inhibition of GluN2D in adult hippocampus can reduce GABAergic interneurons controlled feedback and feed-forward inhibitory circuits, resulting increased hippocampal network activity and synaptic plasticity. Additionally, in mouse models of depression, GluN2D inhibition recovered synaptic excitation-inhibition balance, reversed long-term potentiation deficits, and mimics the cellular and behavioral antidepressant-like actions of ketamine with fewer side effects. Together, this work provides new sights into mechanisms of fast-acting antidepressant actions and suggests GluN2D as a novel potential drug target. The experiments are carefully designed and executed.

We sincerely appreciate the reviewer's positive feedback and thoughtful comments. We have carefully addressed all suggestions, and we believe that the manuscript has been substantially improved through the inclusion of new experiments and targeted revisions. These additions have provided greater clarity and depth to our findings, thereby enhancing the overall quality of the study. We are confident that the revised manuscript now offers a more comprehensive understanding of the role of GluN2D in rapid antidepressant mechanisms. Once again, we thank the reviewer for their valuable input.

Major points:

1. Rodent models just mimic certain aspects of depression at most. Usually conclusions from animal models are summarized as depression-like, or anti-depressant-like. Also before the drug targets confirmed in clinical trials, they are usually termed as potential drug targets.

We agree with the reviewer's suggestion and have revised the manuscript accordingly by consistently using terms such as "depression-like" and "antidepressant-like" when describing findings from rodent models. Furthermore, we now refer to GluN2D as a "potential drug target" (also in the title) to reflect its current status in preclinical research, acknowledging that its therapeutic relevance remains to be confirmed in clinical trials.

2. 10 and 20uM NAB-14 inhibits NMDAR-current from both hippocampal interneuron and pyramidal neurons very similarly (Fig. 1d), why do they have very different effects on the EPSP also measured from hippocampus (Fig.2b)? Could off-target effects other than GluN2D contribute to this?

Thank you for pointing out this lack of clarity. This remark closely resembles Comment 4 from Reviewer #1. We would like to refer the reviewer to our response to that comment and hope that the corresponding additions to the manuscript have helped to provide greater clarity.

While we cannot entirely rule out the possibility of off-target effects, we believe that the explanation outlined above - based on well-established differences in circuit architecture and receptor distribution - offers a more plausible account of the divergent outcomes.

3) NAB-14 and ketamine treatments led to quite different impacts on vGlu1 and GAD65 staining in hippocampus. Is it possible these reflect differences in underlying mechanisms between NAB-14 and ketamine? Since ketamine also led to different impacts as NAB-14 in other experiments, such as Fig.2b-c, it seems like an overstatement to claim that "GluN2D antagonism could fully mimic the cellular and behavioral antidepressant actions of ketamine".

The reviewer is certainly correct in noting that GluN2D antagonism does not *fully mimic the cellular and behavioral antidepressant actions of ketamine*. We have accordingly toned down our statements (now phrased as "*mimic many of...*") and added a paragraph to the Discussion (lines 490–499) explicitly stating that the preferential inhibition of NMDAR-mediated currents in interneurons may represent an important and convergent early signal underlying the downstream behavioral effects of both selective GluN2D inhibition and ketamine. However, ketamine is a "*dirty drug*" with numerous additional targets that may amplify or complement its antidepressant mechanism. In the subsequent sections of the Discussion, we highlight several of these additional pathways.

4) What is the knockdown efficiency in Fig. 4d in the targeted brain regions? The western blot sample images in sFig.5b is very difficult to understand. There are barely any bands. How specific is this antibody?

We acknowledge the reviewer's concern regarding knockdown efficiency in Fig. 4d and the Western blot images in Supplementary Fig. 5b. Indeed, we encountered considerable challenges with the currently available GluN2D antibodies, which are known to exhibit a high degree of non-specific binding. GluN2D is expressed at low levels in adult mice due to its restricted localization in inhibitory interneurons, and the available antibodies often perform poorly within this low expression range.

The bands in the Western blot images are admittedly unclear, reflecting the technical limitations associated with GluN2D detection using current antibody tools. Despite our efforts to improve the signal using multiple alternative antibodies during the revision process, we were unable to obtain reliable results. Consequently, we decided to discontinue this approach due to its lack of clarity and reproducibility and to remove this part of the figure.

Importantly, despite these antibody-related challenges, we obtained strong supporting evidence from alternative experimental approaches. PCR data confirm the downregulation of GluN2D mRNA in the targeted brain regions. Moreover, the behavioral and electrophysiological results, along with the comparisons to scrambled siRNA controls, provide further validation of effective knockdown. The consistency of these findings across multiple independent methodologies strengthens our confidence in the results, despite the lack of protein-level confirmation by Western blot.

We continue to consider the siRNA approach a valuable tool for validating our findings obtained with small molecules.

5) NAB-14 treatment boosts both excitatory and inhibitory (Fig. 5c, measured with GAD65) synaptic markers. Why does the manuscript conclude that “GluN2D inhibition shifts synapses toward excitation (Fig. 5 title)?

We agree with the reviewer’s observation that the conclusion presented in the original manuscript was not fully supported by the data. In the revised version of Fig. 5, we have therefore avoided overinterpretation. We added a new dataset demonstrating an increase in spine density in biocytin-filled pyramidal cells 24 hours after in vivo treatment (Fig. 6a,b). Additionally, Fig. 6 shows a plastic increase in markers of both excitatory and inhibitory boutons 24 hours after treatment, as well as elevated levels of PSD-95 and AMPARs two hours after treatment, all following repeated stress.

To ensure clarity and focus, we refrained from comparing potential differences between NAB-14 and ketamine, as well as between excitatory and inhibitory synapses. Instead, we concentrated on the plastic changes in synaptic and AMPAR density induced by GluN2D inhibition within the RS protocol.

6) Which brain region is responsible for GluN2D inhibition-induced antidepressant-like effects?

We appreciate this important and forward-looking question. Identifying the specific brain regions and circuits through which GluN2D inhibition exerts antidepressant-like effects is indeed of great interest. However, this level of resolution lies beyond the scope of the present study and would require region-specific interventions and circuit-level dissection, which we consider an important goal for future research.

GluN2D-containing NMDA receptors are generally expressed at low levels but are distributed across multiple brain regions, including the hippocampus, prefrontal cortex (PFC), and cerebellum. Among these, the hippocampus and PFC are of particular relevance, as both have been strongly implicated in the pathophysiology of depression and in the mechanisms of antidepressant action.

In the present study, we investigated the cellular mechanism of GluN2D inhibition in the hippocampus, where NAB-14 promotes disinhibition of pyramidal neurons. Similar microcircuit motifs are also present in the PFC. To assess in vivo relevance, we included fiber photometry data

demonstrating NAB-14–induced, activity-dependent disinhibition of pyramidal neurons in the PFC as well (Fig. 3).

Given the systemic administration of NAB-14 and the broad, albeit low, expression of GluN2D, we propose that the observed behavioral effects are likely mediated by circuit-level interactions rather than activity in a single brain region. In particular, interactions between hippocampal and prefrontal networks are known to play a central role in emotional memory and mood regulation. Disentangling the specific regional and pathway-specific contributions will require targeted future approaches. First results from this project dissecting the role of the PFC-Nucleus Reuniens-hippocampus network in antidepressant response are already under review at Nature Comm. and available as preprint: <https://doi.org/10.21203/rs.3.rs-6348176/v1> .

We have added a paragraph (lines 421-434) to the revised manuscript clarifying that the precise neural circuits and brain regions underlying the antidepressant-like effects of NAB-14 remain to be determined and are likely embedded within broader network-level dynamics.

7) The drug concentration and doses are both described using numbers without units throughout the figures. This is confusing.

In response to this comment, we have added the appropriate units for all drug concentrations and doses across the figures to ensure consistency and enhance clarity.

8) Does the pharmacokinetics/pharmacodynamics property of NAB-14 support that inhibition of GluN2D is responsible for the behavioral effects that NAB-14 treatment provides?

In response to this comment, we have now included initial pharmacokinetic (PK) data on plasma and brain concentrations following intravenous and oral administration of NAB-14, and we compare these findings with published PK data for ketamine. Our results demonstrate a short plasma half-life after i.v. administration, comparable to that of ketamine, as well as good brain penetration and moderate oral bioavailability. Furthermore, dose-dependent plasma concentrations were detectable for more than 6 hours (Fig 7a,b and Supplementary Fig. 6a,b). These findings support the notion that GluN2D inhibition may underlie the behavioral effects observed with NAB-14.

9) Is GluN2D required for ketamine's antidepressant like effects?

This is an insightful question that highlights a fundamental challenge in the field. While we fully recognize the importance of determining whether GluN2D is essential for the antidepressant-like effects of ketamine, directly addressing this question poses substantial experimental and conceptual difficulties.

As the reviewer suggests, one potential approach could involve the use of GluN2D knockout mice. However, this strategy has important limitations. GluN2D is highly expressed during early development and plays a critical role in shaping synaptic plasticity and network formation. Constitutive knockout models, in which GluN2D is absent from birth, are therefore prone to widespread compensatory changes that may obscure its specific function in adult antidepressant mechanisms. We discuss this issue in more detail in lines 481-489 of the revised manuscript.

An alternative and more physiologically relevant strategy involves conditional knockout models or acute pharmacological or genetic inhibition in adulthood, thereby circumventing developmental confounds. In this context, we have already conducted experiments using siRNA-mediated knockdown of GluN2D in adult animals, which produced a robust antidepressant-like effect. Based on these findings, we would expect conditional KO models to yield similar results, further supporting the idea that GluN2D inhibition is sufficient to recapitulate aspects of ketamine's action.

Given that GluN2D knockdown alone induces antidepressant-like effects, combining ketamine with GluN2D knockdown or knockout would likely offer limited additional mechanistic insight. Any additive or synergistic effects may be obscured by overlapping downstream consequences, particularly since GluN2D inhibition already shifts excitatory/inhibitory balance and network function in a manner relevant to antidepressant mechanisms.

In summary, while this question is scientifically important, the experimental approaches required to resolve it face substantial limitations. Our data suggest that GluN2D inhibition alone is sufficient to produce antidepressant-like effects, underscoring its role in modulating network plasticity and E/I balance. Given ketamine's pharmacological profile as a "dirty drug" with multiple molecular targets, it is likely that ketamine shares some downstream mechanisms with GluN2D antagonism while also engaging distinct pathways. We have added a paragraph addressing this point in the Discussion (lines 490-499; see also our response to Reviewer #2, comment 3). There, we have added to recent references from independent studies that support an antidepressant action of other GluN2D antagonists.

10) Recent publications suggest that different GluN subtypes lead antidepressant-like effects through different mechanisms. The relationship between GluN2D and other GluN subunits associated antidepressant-like mechanisms should be discussed and better explored experimentally, especially with GluN2B, which also works through a mechanism from inhibitory neurons.

We fully agree that this is a highly relevant point, particularly with respect to GluN2B. In response to this comment, we conducted a series of additional experiments to further investigate the relationship between GluN2D- and GluN2B-mediated signaling.

First, we recorded NMDAR-mediated EPSCs in both pyramidal neurons and interneurons and assessed their sensitivity to the GluN2B-selective antagonist RO25-6981. Interestingly, we did not observe a preferential inhibitory effect on interneurons. If anything, RO25-6981 tended to reduce EPSC amplitudes more strongly in pyramidal neurons (Fig. S1i). We also applied Ifenprodil, a GluN2B-selective NAM at low concentrations that acts as a broader NMDAR antagonist at higher doses. Across a concentration range from 0.1 to 100 μ M, Ifenprodil reduced EPSCs in both cell types without showing any differential effect between interneurons and pyramidal neurons (Fig. S1h).

Second, to evaluate potential overlap in downstream signaling between GluN2B and GluN2D pathways, we performed Western blot analyses (Fig. S5c-f). Acute hippocampal slices were incubated for 45 minutes with NAB-14, RO25-6981, ketamine, or vehicle control. We then assessed phosphorylation levels of ERK1/2 (p42 and p44), key mediators of GluN2B-associated ERK-dependent synaptic plasticity, as well as pAKT. As expected, RO25-6981 increased ERK phosphorylation. In contrast, NAB-14 did not affect ERK phosphorylation but significantly increased pAKT levels, suggesting engagement of a distinct signaling cascade. Ketamine increased both pp42/p44 and pAKT levels.

11) Why aLTP can be induced (Supplementary Figure 3d), while weak-aLPT is inhibited (Fig 3c) by 10uM ket.

We appreciate the reviewer's careful observation. We added two sentences (lines 240-246) with a likely explanation for this finding : "The lack of inhibition of aLTP by 10 μ M ketamine, in contrast to the complete suppression of weak-aLTP (Fig. 4c), can most likely be explained by the approximately 50% reduction in postsynaptic Ca^{2+} influx at this concentration. Under strong stimulation conditions (125 EPSP \rightarrow AP pairings), intracellular Ca^{2+} levels remain sufficient to induce LTP. However, during the weak stimulation protocol (25 EPSP \rightarrow AP pairings), intracellular Ca^{2+} does not reach the critical threshold required for LTP induction."

However, we believe that an experimental validation of this hypothesis falls outside the scope of the present study.

12) Does inhibition of GluN2D change baseline excitatory or inhibitory synaptic transmission?

We thank the reviewer for raising this important point. To address it, we recorded spontaneous EPSPs in naïve animals, after RS and following i.p. application of NAB-14 and KET in RS mice. We observed an increase in the frequency of spontaneous EPSPs, indicating an enhancement of baseline excitatory drive (Fig. 5e).

This effect is consistent with our broader interpretation that GluN2D inhibition results in circuit-level disinhibition, most likely via actions on inhibitory interneurons. The observed increase in spontaneous excitatory activity supports the notion that NAB-14 and KET shift the E/I balance toward a more excitatory state even at rest, prior to any external stimulation or LTP induction protocol.

We believe these findings further support our conclusion that GluN2D inhibition alters synaptic dynamics in a manner that facilitates pyramidal neuron activation under baseline conditions, which may contribute to the behavioral and plasticity-related effects observed in this study.

13) Why compare immobile time of the same CDM mice on day 5 with it on day 8 (Fig 6a), instead of comparing the CDM-Sal mice to drug-treated on day 8?

We thank the reviewer for this suggestion. However, we note that direct between-group comparisons are limited by high inter-individual variability and large standard deviations, making interpretation challenging without substantially larger sample sizes (c.f. Supplementary Fig. 6c).

For this reason, our analysis focuses on within-group comparisons, which reduce baseline-related noise and better capture treatment-related changes over time. Notably, as shown in our review on the RS paradigm (Vestring et al., 2021), immobility in CDM-Sal mice remains stable for several weeks post-stress, supporting the view that reductions in drug-treated animals are unlikely to reflect spontaneous remission.

While we agree that between-group comparisons can be informative, we believe that within-group analyses are more robust under the current design, except in the 48-hour experiment (Fig. 7c), which was randomized, blinded, and parallel-grouped, allowing for reliable group-level comparison.

14) Is SPT initiated 24 hours after siRNA injection in Fig 6b? How fast can siRNA knockdown GluN2D in vivo?

We sincerely apologize for this error in the original manuscript. All assessments following siRNA administration were conducted after a resting period of 72 hours, except for the SPT (Fig. 7d), where 48 hrs elapsed between siRNA application and start of the POST1 period of the SPT (that lasted additional 48 hrs). At this time point, GluN2D mRNA levels were suppressed (Supplementary Fig. 4c), and GluN2D protein was most likely reduced - although the corresponding Western blot was removed from the revised manuscript due to quality concerns. This error has been corrected in the updated version of the manuscript.

15) There is previous literature showed that S-ketamine at dose of 10mg/kg significantly increased travel distance (PMID:26327690), while Fig 7c showed the locomotion of mice was not increased after ket treatment. What makes the difference?

We appreciate the reviewer's careful reading of our manuscript and the effort to relate our findings to existing literature.

In our experiments, we used a racemic mixture of R-/S-ketamine at a total dose of 10 mg/kg (i.e., 5 mg/kg S-ketamine). In contrast, the study referenced by the reviewer examined the effects of pure enantiomers and indeed reported a significant increase in locomotor activity at higher doses (10 and 20 mg/kg). Notably, that study also included a 5 mg/kg S-ketamine condition. However, in their Fig. 5a, there are no indicators of statistical significance for this dose, and the raw data are not provided in sufficient detail to allow for reanalysis.

Based on our interpretation of their Fig. 5a, it appears that 5 mg/kg S-ketamine does not significantly increase locomotor activity in their setup either (R-ketamine had no effect on locomotion at any dose).

This is consistent with our own results: 10 mg/kg racemic ketamine (5 mg/kg each of S- and R-ketamine) did not produce a statistically significant increase in locomotor activity, although we did observe a small numerical increase that did not reach significance (Fig. 8c).

16) Is the GluN2D expression changed in CDM mice of both sex? Also, it will be better to show the data from different genders in different colors.

We agree that a potential modulation of GluN2D expression in animal models of depression or in depressed patients is of significant interest. However, this question remains unresolved, and the available literature is limited. We have clarified this point in the revised manuscript to avoid overinterpretation (lines 96-98).

We would like to emphasize that our study does not claim that GluN2D expression is necessarily increased by stress. While this remains a possibility, we did not directly assess GluN2D expression levels in RS mice. Technically, this is challenging due to the generally low expression of GluN2D in the adult brain. Any potential changes in response to stress are likely to be subtle and would require highly sensitive and specific detection methods. To address this, we are currently preparing a comprehensive mass spectrometry study to examine protein-level changes in the glutamatergic system following RS and antidepressant treatment. However, the results of this resource-intensive study are not yet available.

We also assessed potential sex differences in most of the experimental series included in this study and found no striking effects, as noted in the Methods section. However, our experiments were not powered to systematically detect sex-specific effects, in part due to ethical considerations related to animal use (3R principle) This has now been clearly stated in the Methods section (lines 953-954). For this reason, and to maintain visual clarity, we chose not to display male and female data in separate colors.

Minor points:

17) The mouse inbred line C57BL/6J is widely used. And there were behavioural differences between C57BL/6J and C57BL/6N mice (PMID: 23902802). Why did the authors choose C57BL/6N mice as experimental subjects?

We thank the reviewer for this important comment. We are fully aware of the behavioral differences between C57BL/6J and C57BL/6N mice, as highlighted in the cited study (PMID: 23902802), and we agree that the choice of strain can influence experimental outcomes.

We chose C57BL/6N mice for this study based on our prior experience with stress-based behavioral paradigms in this strain and their broad use in neurobiological and pharmacological research. Although C57BL/6J mice are more frequently employed in social defeat and CUMS protocols (e.g., Golden et al., 2011, Nat. Protoc.), C57BL/6N mice have also been widely used in recent stress-based depression models (e.g., Ye et al., 2024, Eur. J. Pharmacol.; Luo et al., 2025, Front. Neurosci.). These studies demonstrated that C57BL/6N mice reliably develop stress-induced behavioral and molecular phenotypes, including depressive-like behavior and neuroinflammatory changes, supporting their validity for our experimental goals.

In addition, our institutional animal facility maintains well-characterized C57BL/6N breeding lines. To ensure internal consistency across all experimental arms, we therefore continued using C57BL/6N animals throughout this project.

Reviewer #3:

The manuscript by Vestring et al. investigates the role of the GluN2D subunit of NMDA receptors in producing rapid antidepressant effects in mice. By comparing the effects of a GluN2D antagonist NAB-14 with ketamine, the author demonstrated that inhibition of GluN2D reduced NMDAR currents in interneurons but did not affect pyramidal neurons within hippocampal slices. Consistently, inhibiting GluN2D or somatostatin (SOM)-positive interneuron activity facilitated long-term potentiation and increased synaptic protein expression in the hippocampus. Further investigation into the role of GluN2 antagonism using a mouse stress model revealed that GluN2D inhibition restored both synaptic and behavioral deficits induced by repeated stress exposure. Overall, this manuscript is well-written, and the findings are compelling. However, there are concerns about the methods used for GluN2D knockdown in the stress model and a lack of rigor in morphological analysis. I hope the authors will consider addressing the suggestions below to enhance the manuscript further.

We sincerely appreciate the reviewer's thorough evaluation and constructive feedback. The detailed comments and suggestions were highly valuable in refining our manuscript. We have carefully addressed the concerns raised, leading to greater rigor in both experimental design and data interpretation. We believe these revisions have significantly strengthened the manuscript and improved the clarity of our findings. Thank you again for your thoughtful review and for contributing to the improvement of this work.

1. While the authors' decision to conduct GluN2D knockdown in the nervous system to examine its role in stress-induced behaviors is commendable, given the electrophysiological data from hippocampal slices, it would be highly beneficial to investigate the specific effects of GluN2D knockdown on SOM interneurons in the hippocampus. This would offer a more comprehensive understanding of the research findings.

We sincerely appreciate this thoughtful suggestion. We agree that GluN2D likely exerts its effects in a cell type-specific manner and consider SOM interneurons to be strong candidates for mediating GluN2D-dependent disinhibition in the hippocampus. Investigating the precise contribution of GluN2D in SOM cells is indeed a compelling direction for future work.

At the same time, we believe that the behavioral effects observed, particularly those related to emotional processing and antidepressant-like outcomes, are unlikely to result from changes in a single microcircuit or cell population alone. Emotional behavior emerges from the dynamic interaction of local and long-range circuits. For example, disrupting key hubs such as the nucleus reuniens, which links the prefrontal cortex and hippocampus, would likely abolish antidepressant-like effects, even if local hippocampal disinhibition were intact.

This highlights the need to consider both microcircuit and macrocircuit mechanisms in shaping complex behaviors. While GluN2D inhibition within interneuron populations contributes to local disinhibition, as shown in our hippocampal recordings, we propose that such changes must be embedded within a broader network context to produce behavioral effects. Modulating activity in a single region is unlikely to suffice in isolation.

In this study, we demonstrate disinhibition of pyramidal neurons in the hippocampus following NAB-14 treatment and observe a potentially related effect in the prefrontal cortex in newly added fiber photometry experiments (Fig. 3). These findings support a distributed, circuit-level mechanism. Although cell type-specific manipulations (e.g., GluN2D knockdown in SOM interneurons) would be highly informative, they require dedicated tools and were beyond the scope of this study. Our DREADD experiments (Fig. 4e-g) may be seen as an initial step in that direction.

We have clarified in the revised Discussion (lines 421-434) that both local and long-range circuit interactions are likely essential for mediating the antidepressant-like effects of GluN2D inhibition. Please also refer to our response to comment no. 6 from reviewer #2.

2. The authors claim that GluN2D inhibition reverses stress-induced impairment in synaptic morphology and density. However, they have only conducted immunostaining and western blotting of synaptic proteins without directly analyzing synaptic morphology. It would be valuable to assess the density and morphology of dendritic spines to provide clearer insights.

We fully agree with the reviewer that direct analysis of dendritic spine density is important to support our interpretation. To address this point, we conducted additional experiments using biocytin-filled whole-cell recordings to assess spine density in pyramidal neurons. These new data have been included in the revised manuscript (Fig. 6a,b).

We hope this addition addresses the reviewer's concern and provides clearer structural evidence for the effects of GluN2D inhibition.

3. What is the duration of the effect of NAB-14? A notable feature of ketamine's antidepressant action is not only its rapid onset but also its prolonged effects. A comparative analysis would be highly informative.

We agree that the duration of NAB-14's effect is a key aspect, particularly in comparison to KET, whose prolonged antidepressant-like actions are well documented.

To partially address this, we conducted additional behavioral experiments assessing more sustained effects of NAB-14 and KET. We now include new data on escape behavior up to 48 hours after administration (Fig. 7c), as well as results from the nose-poke sucrose preference task, which assessed motivational anhedonia in the 24-48 hour time window (Fig. 7d). These findings provide a clearer picture of the time course of NAB-14's action and allow for a more direct comparison with known features of ketamine's effects.

In addition, initial pharmacokinetic data for NAB-14 (Fig. 7a,b; Supplementary Fig. 6a,b; newly included in the revised manuscript) are comparable to previously reported data for ketamine. The short half-life of both compounds following i.v. administration - substantially shorter than the duration of their behavioral effects - supports the notion that NAB-14 and ketamine may share mechanisms that mediate both rapid and sustained antidepressant-like actions.

That said, important questions remain regarding the optimal dosing frequency, route of administration, and duration of action for the potential therapeutic use of GluN2D NAMs in depression, particularly beyond the 48-hour time frame. These aspects will be explored in ongoing and future studies.

4. How does stress activate GluN2D? Specifically, what mechanisms might underlie the selective recruitment of GluN2D in response to stress?

Although findings in human patients suggest that elevated GluN2D expression may be associated with greater antidepressant responses to ketamine - indicating a potential link between GluN2D, depression, and treatment efficacy - we did not directly investigate this relationship in the present study. Nevertheless, these clinical observations are highly intriguing and further underscore the relevance of GluN2D in mood regulation.

As outlined in our response to comment no. 16 from reviewer #2, we would also like to clarify that we do not claim that stress directly activates or increases GluN2D expression, nor do we suggest that GluN2D is selectively recruited in response to stress. While this is an interesting and testable hypothesis, our study did not examine whether GluN2D levels change in stressed animals.

To address this question, we are currently preparing a comprehensive mass spectrometry study to investigate protein-level changes in the glutamatergic system following repeated stress and

antidepressant treatment. However, results from this resource-intensive effort are not yet available.

We have clarified this point in the revised manuscript to avoid overinterpretation (lines 96–98).

5. In Fig. 5e and 5f, the values are normalized to those of CDM. Why were actual values not used for comparison? Is this normalization intended to account for experimental variations? It would be beneficial to normalize against the values of naïve mice to illustrate the variability observed under CDM.

We thank the reviewer for this helpful suggestion. In the original manuscript (Fig. 5e,f; now Fig. 6g,h), we normalized values to the CDM group to emphasize treatment-induced changes relative to the stressed condition. However, we agree that normalization to naïve controls provides important additional context for evaluating the variability introduced by the CDM protocol.

In response, we have performed the suggested normalization against naïve animals. These results are now included in the revised manuscript as Supplementary Figure 5a,b. We hope this additional analysis helps to clarify the range and direction of effects observed across experimental groups.

6. Regarding Fig. 7c and 7d, how much time elapsed between drug administration and the OFT? It is known that a single ketamine administration produces anxiolytic effects 24 hours post-administration.

We observed an anxiogenic effect one hour after drug administration, which is consistent with findings in the literature showing that ketamine can produce anxiety-like behaviors in the acute phase following treatment. For instance, Acevedo et al. found a decreased time spent in the center of an Open Field 30 minutes after ketamine administration, although at a slightly higher dosage (PMID: 36496189). However, it is important to note that the data regarding ketamine's acute anxiogenic effects are heterogeneous, and there is limited consensus in the literature. Some studies report no significant changes or even contradictory results in this early phase, making the acute effects less well-documented.

In contrast, the anxiolytic effects of ketamine are well-established at later time points, typically around 24 hours post-administration. For example, Zhang et al. (PMID: 25231918) demonstrated that ketamine reliably reduces anxiety-like behaviors in animal models of posttraumatic stress disorder at this time period, aligning with its known antidepressant properties. We cited these studies in the revised manuscript (lines 379-381).

In this study, we specifically used the OFT at an earlier time point (1 hour post-administration) to assess the potential side effects of NAB-14, selecting a window where ketamine's effects are known to be more variable and may include anxiogenic responses. While we acknowledge that ketamine generally produces anxiolytic effects after a longer duration, this short-term assessment allows us to focus on the acute side effect profile, which is critical for evaluating new compounds like NAB-14.

7. The western blot signals in Supplementary Fig. 5b could be clearer. The author should present the entire membrane or indicate molecular size alongside the membranes.

For this point, we would like to refer the reviewer to our response to comment no. 4 from reviewer #2 in this document.

8. Could the authors specify the post-hoc tests used in this study not only in the Supplementary Table but also in the Figure legends? The layout of Supplementary Table 1 in the PDF appears distorted.

In response to the request for greater clarity regarding the statistical analyses, we have now specified the post hoc tests in the figure legends.

We apologize for the formatting issue with Supplementary Table 1 in the PDF. To address this, we have incorporated all relevant statistical information into the figure legends wherever possible and have uploaded the complete source data, including detailed descriptions of the statistical methods, to Figshare (see box at the beginning of this point-to-point response for private links), in accordance with the guidelines of *Nature Communications*.

9. Fig .6a is not cited anywhere in the manuscript. Please include a citation for this figure.

We apologize for the error concerning Fig. 6a in the original manuscript. Due to the inclusion of newly added experimental series, the order of some figures has changed. We have ensured that all figure citations in the text are now correct and complete.

Self-Initiated Changes Between Initial Submission and Revised Version

1. Addition of Co-Authors

We have added the following co-authors who contributed additional experiments for the revised version:

- Shira Knafo, Magdalena Wojitas: Analysis of biocytin-filled cells for synaptic morphology (Fig. 4e)
- Jean Paul Grohe: Biocytin fillings (Fig. 4e) and spontaneous EPSP recordings (Fig. 4f)
- Jan Warneke: Recordings of GluN2B EPSCs and additional experiments with NAB-14 in interneurons and pyramidal cells
- Jule Wendel: Additional Western blot experiments (Fig. S1j)
- Louise Ellen Schuberth: Analysis of additional data sets for the revision

2. Correction in Abstract

In line 37 of the abstract, we corrected “selective GluN2D inhibitors” to “the selective GluN2C/D inhibitor NAB-14.”

3. Terminology Update

We consistently replaced the term “chronic despair model (CDM)” with “repeated stress (RS)” throughout the manuscript to avoid anthropocentric language.

4. Updated References

We added two recent references (PMID: 40043126 and 38779864, lines XX–XX) supporting a potential role of GluN2D in antidepressant response. These publications from other groups were published after the initial submission of our manuscript and further substantiate our hypotheses.

5. Statistics update

We updated all statistics and corrected some errors in the figures and their legends.

Vestring et al.:

**The NMDA receptor subunit GluN2D is a potential target for rapid antidepressant action
(NCOMMS-24-46210)**

Response to the reviewers' comments

Reviewer #2 (Remarks to the Author):

The authors have provided adequate explanations in the rebuttal letter or changes in the manuscript to address most of my concerns.

Reviewer #3 (Remarks to the Author):

The authors appear to have appropriately addressed most of the concerns raised by this reviewer. However, the reviewer would like to seek clarification regarding the authors' interpretation of the neuronal morphological analysis presented in the newly added Fig. 6a and 6b. While GluN2D inhibition resulted in an increase in spine density, consistent with the authors' overall claims, repeated stress did not appear to reduce spine density. This finding seems to contrast with the observed decrease in synaptic protein expression and spontaneous EPSC frequency following repeated stress. The reviewer would appreciate it if the authors could elaborate on this apparent discrepancy, for instance, whether the intensity or duration of the stress protocol employed in this study may have been insufficient to elicit detectable changes in spine density.

We thank the reviewers for their thoughtful and constructive feedback and their support of our work. Regarding the remark of reviewer #3, we fully agree with his notion and added an additional sentence in lines 297-299: "Although repeated stress did not significantly reduce spine density, this likely reflects the moderate and chronic nature of the applied paradigm, which induces subtle but functionally relevant synaptic alterations rather than pronounced structural remodeling."